# Protracted development of gaze behaviour

Marcel Linka ✉, Harun Karimpur & Benjamin de Haas

How does scene viewing develop? Previous evidence is limited and suggests that viewing behaviour may be adult-like from about eight years old. Here we present data from $n$ = 6,720 participants from 5 to 72 years old, freely viewing 40 natural scenes. We found that the development of scene viewing is surprisingly protracted. Semantic salience for social features continuously changes until adolescence, and text salience increases over the first two decades of life. Basic oculomotor biases towards the image centre and along the horizontal meridian develop until adolescence, matching developmental changes in visual sensitivity and cortex. Finally, while the tendency for visual exploration continuously increases, fixation patterns become less idiosyncratic and more canonical throughout adolescence. These findings show that fundamental aspects of adult gaze take up to two decades of continuous development and push individuals towards more canonical viewing patterns. We suggest that development is key to understanding the general mechanisms of active vision.

Visual attention towards natural scenes is drawn to low-level features such as luminance, colour and orientation contrast[1,2], as well as to higher-level features such as objects[3] and their meaning[4]. When predicting free-viewing behaviour, higher-level features such as faces, text, touched objects and taste[5,6] carry more weight than lower-level features[3,5]. The extraction of global meaning is probably a main objective of scene viewing without additional task demands[7–9]. Besides bottom-up stimulus drives, viewing behaviour can be shaped by top-down task demands as well as by internal states[10–12] and traits[13–16] of the observer. Adult active vision is further marked by spatial biases, such as a tendency to fixate central image regions[17–19] and a predominance of horizontal saccades[20,21], along with a corresponding horizontal (and lower visual field) advantage in visual sensitivity[22–26].

A central question in visual perception is whether and how oculomotor biases and attentional preferences develop. Infants[27] and children[28] watching movie clips exhibit greater inter-individual variability than adults, suggesting that common attentional biases are learned. Although gaze preferences for faces and social information are present in infants[29–33] and early childhood[27,34], it remains poorly understood how they evolve from thereon. Amso et al. have reported the development of gaze towards faces to plateau around age eight[35]. We recently found lower text salience and higher salience of hands and objects being touched in preschoolers than in adults[36]. More basic patterns of oculomotor behaviour also seem to

develop during childhood. Saccade amplitudes are higher and fixation durations are shorter at later stages of childhood (ages four to eight) than at age two[37], and the tendency to fixate on central image regions is weaker at age eight than at age four[38]. The horizontal bias seems to be present in infants, albeit probably to a lesser extent than in adults[39]. While most studies report that basic oculomotor behaviour is developing in children, other research finds no substantial differences in visual exploration and fixation frequency between children and young adults[40]. Furthermore, while adult females show enhanced processing of some social stimuli[41–43], this does not seem to translate to generally enhanced face salience[44], and it is unclear whether there are relevant potential gender differences across development. Together, these studies suggest that basic visuospatial and semantic salience biases differ between young children and adults but may become adult-like as early as age eight. Overall, however, our understanding of gaze development is severely limited by a scarcity of data, with most studies comparing two or three age groups with limited sample sizes, which rarely cover adolescence and do not allow a continuous comparison of gaze across development. It thus remains unclear whether and how scene viewing changes beyond early childhood.

At the same time, recent studies have demonstrated that visual field asymmetries typical for adults are comparatively diminished in both children and adolescents. Specifically, adults show a stronger

Experimental Psychology, Justus Liebig University Giessen, Giessen, Germany. ✉e-mail: marcellinka54@gmail.com

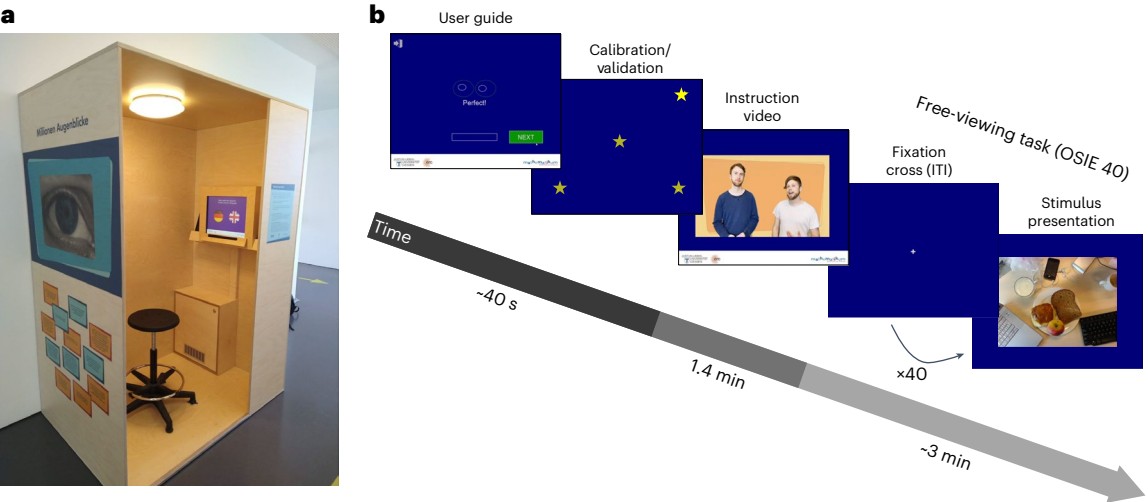

**Fig. 1 | Eye-tracking booth and procedure. a,** The eye-tracking booth exhibited at Mathematikum. **b,** Overview of the experimental procedure (ITI refers to inter-trial interval). For a more detailed description of the software, please refer to Supplementary Fig. 1.

enhancement of visual performance along the horizontal than along the vertical meridian and for the lower than for the upper vertical meridian[45,46]. The diminished lower visual field advantage in adolescents goes along with a more balanced V1 representation of the lower and upper vertical meridian in terms of surface area[47]. In the ventral stream, cortical preferences for highly salient stimulus categories continue to develop into adolescence, along with corresponding recognition skills[48–50]. Moreover, recent work has shown that the success of salience model predictions for scene viewing can hinge on the age of participants[51]. Together, these studies suggest that basic aspects of visual processing continue to evolve into adolescence and raise the possibility that this extends to oculomotor biases and attentional preferences.

Here we aimed to test whether scene viewing matures early or develops into adolescence and beyond. We therefore traced the developmental trajectories of semantic salience and basic oculomotor biases continuously and across a wide age range. The size of the present dataset further allowed us to investigate gender differences in semantic salience and their potential variation across age. In collaboration with a local science museum, we gathered gaze data from over 10,000 participants ranging from 5 to 72 years old. Of these, we selected 6,720 participants for further analysis on the basis of the high quality of their data. The participants sat in an eye-tracking booth and freely viewed 40 everyday scenes (see Fig. 1 for an overview), which were carefully selected to maximize sensitivity for individual gaze biases[14] and contained hundreds of semantically annotated objects. To probe the development of semantic salience, we calculated the proportion of dwell time and of first fixations after image onset, directed towards objects of several semantic features. We plotted these averages individually for age bins spanning two years each, ranging from 5–6 to 71–72 years. We then fitted polynomial developmental trajectories across the bin-wise distributions, up until age 57. Our dataset is particularly sensitive to the period from ages 5 to 57, with >90 observers per age bin (Methods and Fig. 2). Similarly, we traced the developmental trajectories of the central and horizontal biases, visual exploration, fixation frequency and inter-observer similarities. We found strong evidence for protracted development of attentional preferences and basic oculomotor biases extending into adolescence and young adulthood. Furthermore, adolescents viewed scenes in more idiosyncratic ways, which gradually gave way to more canonical patterns; these patterns fully matured in the mid-20s.

## Results

### Protracted development of semantic salience

To trace the development of semantic salience, we calculated the proportion of dwell time and of first fixations towards four types of targets for each age bin: text, faces, touched objects and bodies. We then applied polynomial curve fitting and the Akaike information criterion (AIC) to determine the best-fitting developmental trajectories. Specifically, we fitted polynomials of degrees ranging from zero to ten to the data for each metric. For these candidate models, we calculated the AIC differences using the formula $\Delta i = AIC_i - AIC_{min}$, where $AIC_i$ is the AIC value of the $i$th model, and $AIC_{min}$ is the lowest AIC value among all models. We then selected the least complex model with $\Delta i < 4$ as the best fit.

To differentiate periods of broad linear change from stability, we further applied a sliding window to continuously assess the evidence for linear slopes across five contiguous age bins. Figure 3 shows the developmental trajectories of semantic salience for dwell time (Fig. 3a,c) and for first fixations (Fig. 3b,d), as well as pixel masks and the fixations of two age groups for an example image (Fig. 3e).

Text salience steeply and continuously increased, with dwell time and first fixation proportions on text more than doubling between ages 5–6 and ages 15–16 (from about 6% to 17% for dwell time and 4% to 10% for first fixations). For dwell time, this was followed by a shallower increase before finally stabilizing in the 20s. The sliding-widow analysis yielded ΔAIC > 5 for a positive slope until at least age 21–22 for dwell time and ΔAIC > 5 until at least age 15–16 for first fixations. The corresponding developmental trajectories were best described by an eighth-degree polynomial for dwell time and a fourth-degree polynomial for first fixations (see above for details on model selection). These findings proved robust across images with different types of text elements and across participants who indicated German or English as their preferred language, suggesting they are largely independent of language proficiency (Supplementary Information and Supplementary Fig. 4).

Face salience showed a differentiated developmental pattern, with dwell time proportions continuously dropping from ages 5–6 to at least 11–12 (from 42% to 38%; ΔAIC for decrease, >5), whereas the proportion of first fixations on faces increased from ages 5–6 to at least 13–14 (from 45% to 53%; ΔAIC for increase, >5) and more slightly from 15–16 to at least 17–18 (from 53% to 54%; ΔAIC for increase, >5). The corresponding developmental trajectories were best fitted by eighth- and fourth-degree polynomials, respectively.

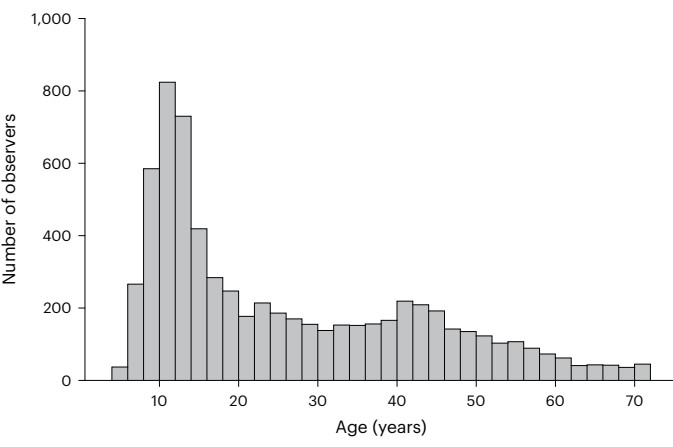

**Fig. 2 | Age distribution of the analysed free-viewing data.** Histogram displaying the distribution of ages of included observers using a bin width of two.

The salience of touched objects continuously decreased, with dwell times dropping from ages 5–6 to at least 13–14 (from 24% to 22%; ΔAIC for decrease, >25) and the proportion of first fixations dropping from ages 5–6 to at least 13–14 (from 21% to 16%; ΔAIC for decrease, >25) and more slightly between ages 15–16 and 17–18 (from 16.5% to 15.6%; ΔAIC for decrease, >5). Furthermore, for first fixations, there was a slight increase between ages 35–36 and 41–42 (from 15% to 17%; ΔAIC for decrease, >5). The developmental trajectories were best described by eighth- and seventh-degree polynomials, respectively.

The salience of bodies showed only slight developmental changes, with one period of slight change in dwell time between ages 9–10 and 11–12 as well as between 15–16 and 17–18 (ΔAIC for decrease, >5). For first fixations there was one period of slight change between ages 5–6 and 9–20 as well as between 35–36 and 39–40 (ΔAIC for decrease, >5). Overall, the developmental trajectory of dwell time proportions was best fit by a fifth-degree polynomial, while that of first fixations was best fit by a third-degree polynomial.

### Protracted development of gender differences
Previous social cognition research suggests that females show a greater visual preference for social stimuli than male participants[41,42,52]. Here we explored gender differences in scene viewing. Surprisingly, males exhibited more dwell time on faces in the 11–16 age range and more first fixations on faces at ages 11–12 and 15–16, while females spent more dwell time on touched objects at ages 5–6, 11–12 and 15–16 and more first fixations at ages 69–70. Pooled across age bins, males showed slightly higher gaze proportions towards faces and females towards touched objects, bodies and text (see Supplementary Information and Supplementary Fig. 5 for more details).

### Protracted development of the horizontal bias
Adults make far more saccades along the horizontal than the vertical[20,21], which matches anisotropies in sensitivity across the visual field[22,23,26]. Age-specific polar histograms (Fig. 4a) and the developmental trajectory of the proportion of horizontal saccades (<45 degrees from the horizontal meridian; Fig. 4b) revealed a strong and prolonged increase in the horizontal bias. The proportion of horizontal saccades increased from 53% to 56% from ages 5–6 to at least 13–14 (ΔAIC > 5), and the best-fitting developmental trajectory (ninth-degree polynomial) reached a plateau from about age 15–16 (see above for details on model selection).

The parallel protracted development of the horizontal bias and text salience raises the possibility of a causal connection between the two. Reading requires mostly horizontal saccades and this acquired behaviour may generalize to scene viewing. We first confirmed that the developmental pattern we observed for the horizontal bias generalizes to images without text elements (Supplementary Results and Supplementary Fig. 6). We then tested the partial correlation of individual differences in the horizontal bias and text salience, which was statistically significant but negligible when controlling for age ($r_{6718} = 0.09$; $P < 0.001$; 95% confidence interval, (0.06, 0.11)) or log-transformed age ($r_{6718} = 0.12$; $P < 0.001$; 95% confidence interval, (0.1, 0.14)). That is, the data did not support the hypothesis of a strong relationship between the horizontal bias and individual text salience beyond their similar developmental trajectories.

### Protracted development of centre bias and visual exploration
Next, we examined developmental changes in the centre bias, as indicated by the mean eccentricity of fixations from the image centroid (Fig. 5a), as well as changes in visual exploration, as indicated by the absolute number of objects fixated (Fig. 5b) and fixation frequency (Fig. 5c).

The development of average fixation eccentricity from the image centre was best fitted by an eighth-degree polynomial. Average eccentricity increased from ages 5–6 to at least 15–16 (from 208 to 246 pixels; ΔAIC > 5) before it slowly decreased from about ages 43–44 to at least 47–48 (from 251.4 to 251.3 pixels; ΔAIC > 5). The development of fixation frequency was best fit by a fifth-degree polynomial (see above for details on model selection). It increased from ages 5–6 to at least 15–16 (from 2.01 to 2.39 Hz; ΔAIC > 25) and more slightly from ages 17–18 to at least 21–22 (from 2.39 to 2.41 Hz; ΔAIC > 5). Furthermore, there was a decrease in fixation frequency from ages 47–48 (2.47 Hz), which coincides with significant evidence for a broad decline at least until ages 49–50 (2.42 Hz; ΔAIC > 5). Overall, estimated fixation frequencies were conservative, given that the event classification procedure could label fixations separated by very small saccades as single fixations (see Supplementary Fig. 3c for examples). Reassuringly, the developmental trend for increasing exploration replicated in the number of fixated objects across all images, which increased from ages 5–6 to at least 15–16 (from 107 to 142; ΔAIC > 15) and more slightly between ages 17–18 and at least 21–22 (ΔAIC > 5). Visual exploration then slowly decreased between ages 43–44 and at least 49–50 (from 147 to 145; ΔAIC > 5) and followed developmental trajectories best fitted by a broadly U-shaped sixth-degree polynomial.

The sliding window analyses using a window size of five age bins yielded strong evidence for the increase in all indicators of visual exploration over the first 15 years of life, but not for the later decline. However, a supplementary analysis using a larger window size and including bins at higher ages with smaller samples and lower sensitivity confirmed significant changes for both periods (Supplementary Information and Supplementary Fig. 7). This reflects the more protracted and gradual decline of visual exploration with older age, compared with the steeper increase early in life.

### Protracted development of entropy and observer similarity
Finally, we traced the development of fixation entropy and individual differences in gaze behaviour. Adults show systematic individual differences in gaze behaviour towards complex natural scenes[13,14]. Here we first computed the Shannon entropy of smoothed fixation maps for each image and age group. This group-level entropy showed a steep early increase and peaked at age 11–12 (ΔAIC > 5), after which it decreased until age 19–20 (ΔAIC > 5). After that, it increased slightly between ages 33–34 and 35–36 (ΔAIC > 5), followed by a second, more prolonged decrease (ΔAIC > 5 between 43–44 and 47–48). At age 71–72, group-level entropy was at its lowest, 39% below its peak. This developmental trajectory was best described by a tenth-degree polynomial (see above for details on model selection; Fig. 6a; see the markers at the top of the figure for periods of significant linear change in the sliding window analysis).

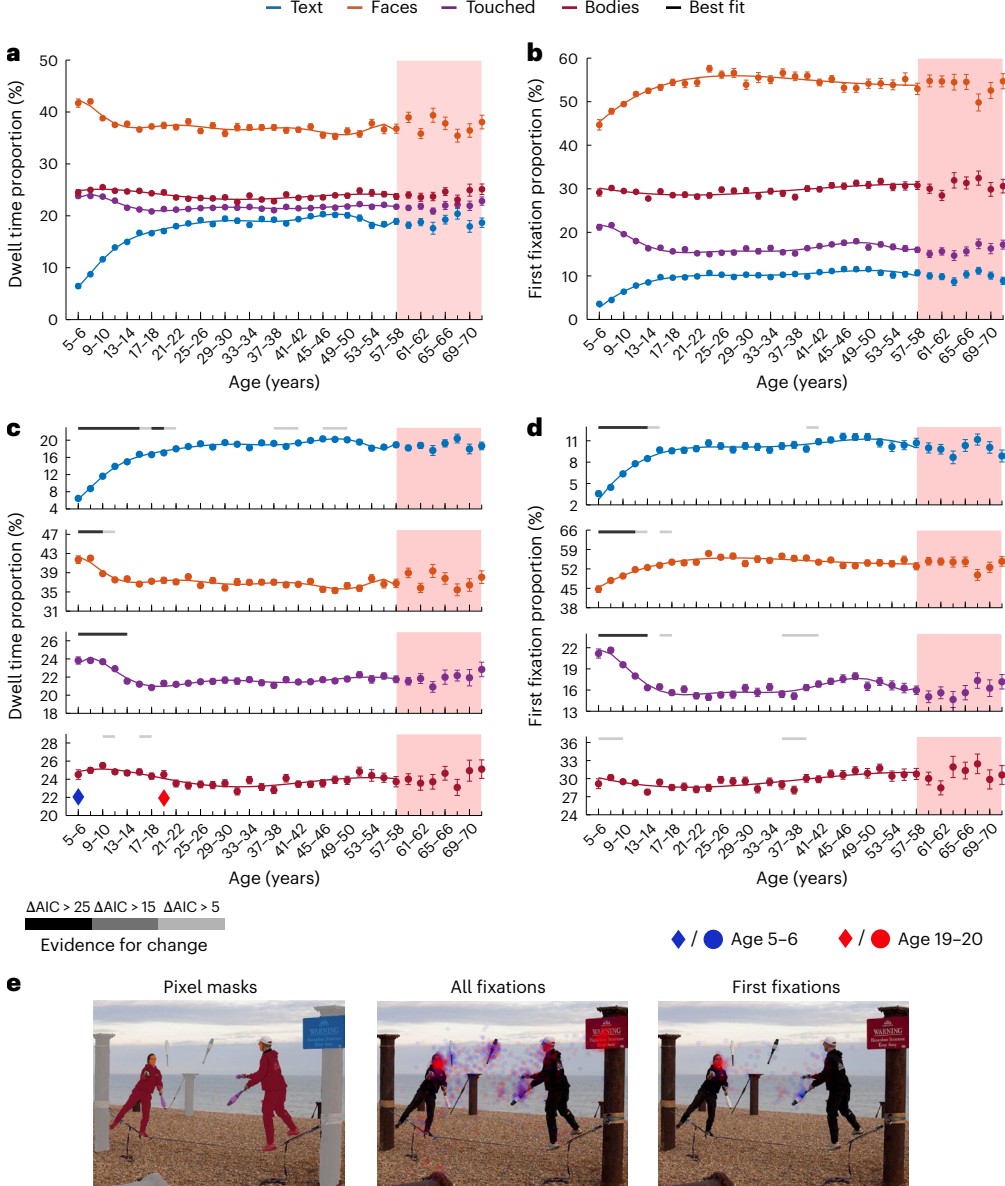

**Fig. 3 | Protracted development of semantic salience. a–d**, The points display the mean proportion of dwell time and first fixations directed towards objects of four semantic features. The error bars indicate ±1 s.e.m. Please note that each age bin spans two years. Due to space constraints, only every second tick mark is labelled. A total of 6,720 participants were included in the analysis; for a more detailed distribution of participants across age groups, please refer to Fig. 2. Semantic categories are indicated by colours, as shown in the key. The lines show the corresponding best-fitting polynomials. In **c** and **d**, the horizontal lines above the plots indicate the level of evidence for a given bin to be part of a broad linear change via shades of grey, as shown in the key (sliding window analysis; see main text and Methods). Red shading marks age bins with lower sensitivity to developmental changes, which were disregarded for curve fitting

(*n* < 90 per bin). Panels **a** and **b** show the trajectories for all features in a single plot for comparisons of scale; **c** and **d** zoom in on each semantic category to highlight developmental changes. **e**, The left image displays object pixel masks for an example image, with colours corresponding to the categories in **a–d** (objects outside these categories shown in light grey). The middle and right images display alpha-blended fixation overlays for the same image, showing all fixations (middle) or only first fixations after image onset (right) for two age groups. Fixation overlays shown in blue and red correspond to ages 5–6 and 19–20, respectively. These age groups are marked with blue and red diamonds above the *x* axis in **c**. Panel **e** adapted from ref. 36 under a Creative Commons license CC BY 4.0.

Group-level entropy reflects both intra-individual fixation dispersion and inter-individual differences. To disentangle the two, we first computed the intra-individual entropy of individual fixation maps and averaged it for each age group (Fig. 6c). Echoing our findings on visual exploration, intra-individual entropy first increased until at least age 15–16 and between ages 17–18 and 19–20 before the best-fitting seventh-degree polynomial stabilized and steadily declined again. Strong evidence of linear change in the sliding window analysis (ΔAIC > 15) was limited to the steepest phase of the initial rise between ages 5–6 and 15–16. A slight decrease was shown between

ages 43–44 and 49–50 (ΔAIC > 5) when using a window size of five bins and extended to a steady decline when using a larger window size of ten bins (Supplementary Results and Supplementary Fig. 7). Like the other indicators of individual visual exploration, this reflects a more protracted and gradual decline in older age, compared with the steep increase early in life (see above).

Finally, to pinpoint the development of inter-individual similarity specifically, we directly compared pairs of observers regarding their individual distributions of dwell time across objects in the images. Specifically, we quantified the average pairwise correlation of object

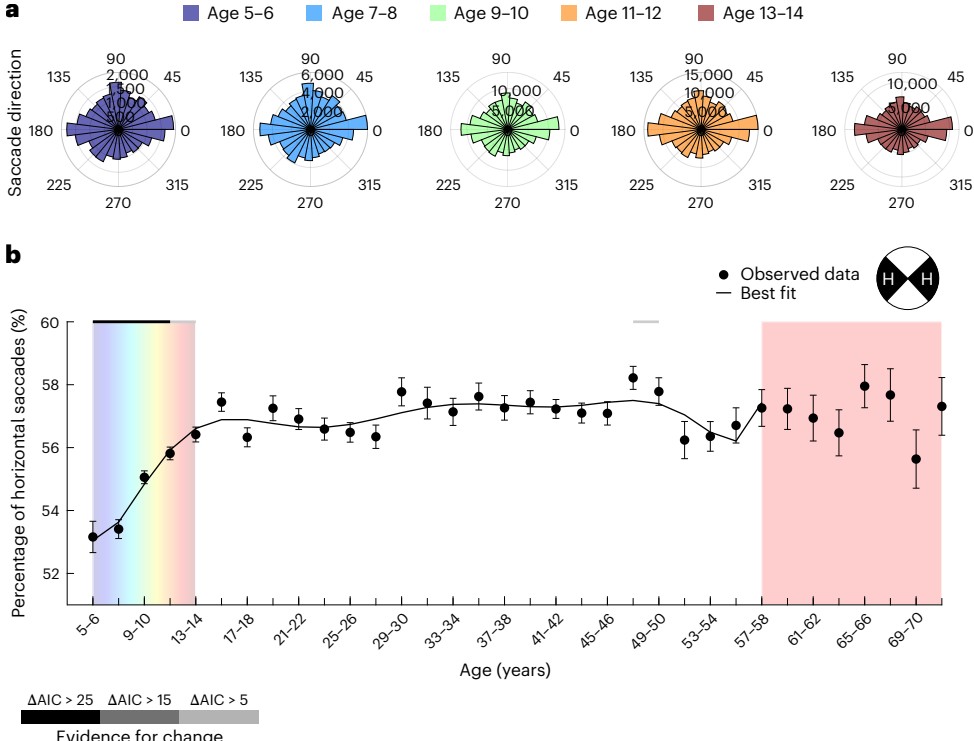

**Fig. 4 | Protracted development of the horizontal bias. a**, Polar histograms for saccades across 36 direction bins for ages 5–6 to 13–14. Age is indicated by colour as shown in the key. **b**, The average proportion of horizontal fixations across age bins ±1 s.e.m. (points and error bars, respectively). $n = 6,720$ participants were included in the analysis; for a more detailed distribution of participants across age groups, please refer to Fig. 2. Please note that each age bin spans two years.

Due to space constraints, only every second tick mark is labelled. The black line shows the best-fitting polynomial. The horizontal lines at the top indicate the level of evidence for a given bin to be part of a broad linear change via shades of grey, as shown in the key (sliding window analysis; see main text and Methods). The red shading marks age bins with lower sensitivity for developmental changes, which were disregarded for curve fitting ($n < 90$ per bin).

fixation patterns for each age bin. As shown in Fig. 6d, pairwise observer similarities decreased from $r = 0.65$ to $r = 0.58$ from ages 5–6 to 13–14, before steeply increasing to $r = 0.63$ at age 23–24 ($\Delta$AIC > 25) onwards. This developmental trajectory was best described by a tenth-degree polynomial.

Taken together, visual exploration and intra-individual fixation dispersion steeply increased until adolescence. This increase in individual exploration went along with a drop in observer similarity. From age 13–14, the distribution of gaze became more canonical across observers, despite individual exploration tendencies remaining at their highest level.

## Discussion

To probe the developmental trajectory of free-viewing behaviour towards natural scenes, we analysed the gaze of thousands of individuals from 5 to 72 years old. Previous studies using small and focused samples have reported differences in semantic salience[27,35,36] and facets of oculomotor behaviour[37,38] between adults and children younger than eight years. Here we found surprisingly protracted development for the semantic salience of text, faces and touched objects, as well as for visual exploration and basic oculomotor biases, such as the central and horizontal biases, all of which continuously developed into adolescence and for some measures up until young adulthood. Inter-observer similarity steeply increased throughout adolescence and plateaued in the mid-20s. This suggests that typical adult gaze behaviour takes two decades to unfold and requires prolonged visual experience. The protracted development of basic oculomotor and high-level biases appears to ultimately lead to more canonical ways of exploring visual scenes. In the following, we discuss each of our findings in detail and the hypothesis that protracted

gaze development reflects a gradual accumulation of priors about the visual world.

### Protracted development of semantic salience

We found a continuous increase in the proportions of dwell time and first fixations directed towards text, which plateaued in the third decade of life. Previous results have shown drastically enhanced text salience in adults compared with preschoolers[36]. Here we found that the development of text salience extends far beyond reading acquisition and takes ~20 years of development. Control analyses focusing on a subsample of participants indicating English as their preferred language or focusing on an image subset with text elements common in German suggest that this effect is unrelated to language proficiency (Supplementary Information and Supplementary Fig. 4). Adult text salience appears to unfold across most of the educational history and may be mediated by levels of literacy and reading experience. An increasing share of text in the foveal diet may have a self-reinforcing effect on text salience. Future work could test this hypothesis directly, using literacy tests and mobile eye-tracking to estimate changes in the visual diet and in gaze behaviour longitudinally. Similarly, longitudinal studies combining functional neuroimaging and eye-tracking could probe a potential bidirectional relationship between the development of text salience and word-preferring patches of cortex in the ventral stream[13,36,48]. Cross-sectional and longitudinal studies have shown that word-preferring ventral regions develop from reading acquisition until at least the age of 17 (refs. 50,53,54).

The salience of objects being touched continuously decreased until adolescence. We speculate that this may be related to an increasing understanding of (inter)actions among people and objects.

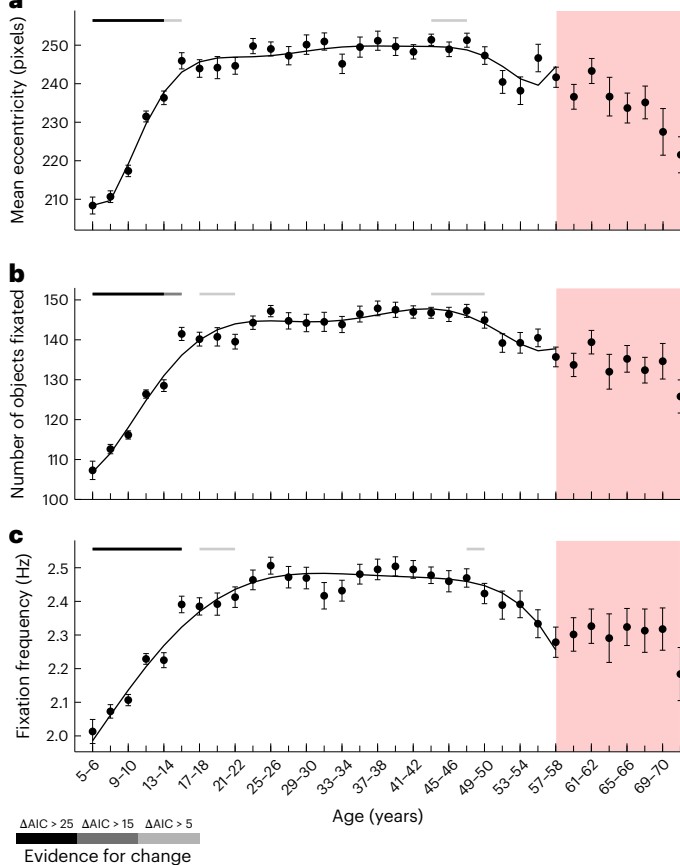

**Fig. 5 | Protracted development of visual exploration. a–c**, The points indicate the mean distance of fixations to the image centroid (in pixels; **a**), the average number of objects fixated (**b**) and the average fixation frequency (Hz; **c**) for age bins ranging from ages 5–6 to 71–72 (x axes). In total, 6,720 participants were included in the analysis; for a more detailed distribution of participants across age groups, please refer to Fig. 2. Please note that each age bin spans two years. Due to space constraints, only every second tick mark is labelled. The error bars indicate ±1 s.e.m. The black lines show the best-fitting polynomials. The horizontal lines above each plot indicate the level of evidence for a given bin to be part of a broad linear change via shades of grey, as shown in the key (sliding window analysis; see main text and Methods). The red shading marks age bins with lower sensitivity to developmental changes, which were disregarded for curve fitting (n < 90 per bin).

Accumulated experience with scenes and actions (for example, two people playing basketball) may enable adults to categorize a given action from scene gist[55], whereas children and adolescents may have a greater need to foveate hands and the things they do for action understanding[56]. This hypothesis could be tested in future studies comparing action understanding from scene gist across different age groups and levels of expertise. Interestingly, a recent longitudinal study revealed that after reading onset, limb-preferring cortical regions shift their preference towards text (and faces). This shift lasts until ages 15–17, suggesting a prolonged process of cortical recycling[48,50], which may be related to corresponding changes in visual diet. The longitudinal neuroimaging studies suggested above could track the push–pull relationship between preferences for hands and text in both gaze behaviour and cortical tuning to test whether they play out in parallel or one precedes the other.

Face salience continuously developed until age 14, a relevant finding for a long-standing debate focusing on its potentially innate nature and very early development[57,58]. Interestingly, face salience showed a differential developmental pattern, with a protracted decrease of dwell time on faces and a parallel increase of first fixations directed towards faces. This finding is consistent with previous work comparing preschoolers and adults[36]. While the individual proportions of first fixations and dwell time on faces were highly correlated with each other in all age groups, their opposing developmental trajectories may point to dissociable contributing mechanisms. The tendency to saccade towards faces immediately after image onset appears stronger and more mature in adults and may primarily be driven bottom-up. A second, top-down mechanism may guide dwell time towards faces and create stronger competition with text salience and visual exploration in adults. The proportion of adult dwell time on text is considerably higher than the proportion of first fixations on text (~19% and ~11%, respectively), indicating a stronger competition of text for dwell time than for first saccades. As mentioned above, recent developmental studies suggest that ventral cortical territory selective for limbs may be recycled for both text and face preferences[48,50]. Future longitudinal research could track and test a relationship between the development of face salience and ultrarapid saccades towards faces[59–61], which presumably present bottom-up mechanisms. Future studies may also test whether the neural signature of immediate and short-latency saccades towards faces is different from that of later and more prolonged fixations towards faces.

For body salience (beyond faces and hands), we found a very subtle developmental change for dwell time across the whole age range tested, but no strong developmental changes. This matches studies finding no changes in cortical activation for bodies along the ventral stream throughout adolescence and young adulthood[50,62], underscoring the dissociable and special nature of faces and hands for the visual system and its development.

### Protracted development of gender differences
We observed gender differences that pointed in the opposite direction of some previous studies[41,42], with males showing slightly stronger face salience and females slightly stronger salience for touched objects, bodies and text (here only for first fixations). However, these differences were most pronounced in younger children and mostly disappeared after adolescence. Pooled across age groups, they had small effect sizes (all d < 0.25), suggesting limited practical significance (Supplementary Information and Supplementary Fig. 5c). Nevertheless, these findings from our very large sample caution against the overgeneralization of gender differences observed in small samples. Additionally, the variation with age shows that gender differences are modulated by further observer characteristics. Future work may probe whether culture is one of these.

### Protracted development of exploration and anisotropies
Previous research has indicated a more pronounced central bias and reduced visual exploration in children aged four compared with older children[38], as well as a less prominent horizontal bias in infants[39]. These studies show that visuospatial biases change during early childhood, but they also suggested their maturation by age eight. Here we found that visuospatial biases develop into adolescence. Specifically, we observed a continuous reduction of the central bias until at least age 15–16 and a comparably protracted increase in the horizontal bias, which was independent of text salience. These protracted changes in saccadic anisotropies match recent findings on visual field geometry. Children and adolescents show a lower sensitivity advantage for the horizontal meridian than adults[45,46]. Future longitudinal studies could disentangle the relationship between the protracted development of the saccadic horizontal bias and visual field anisotropies, which may both be related to the formation of spatial priors for typical scene layouts[20]. Additionally, the visual field distribution of informative features may vary with age. Small children probably encounter relevant stimuli in their upper visual field more often than adults do[46].

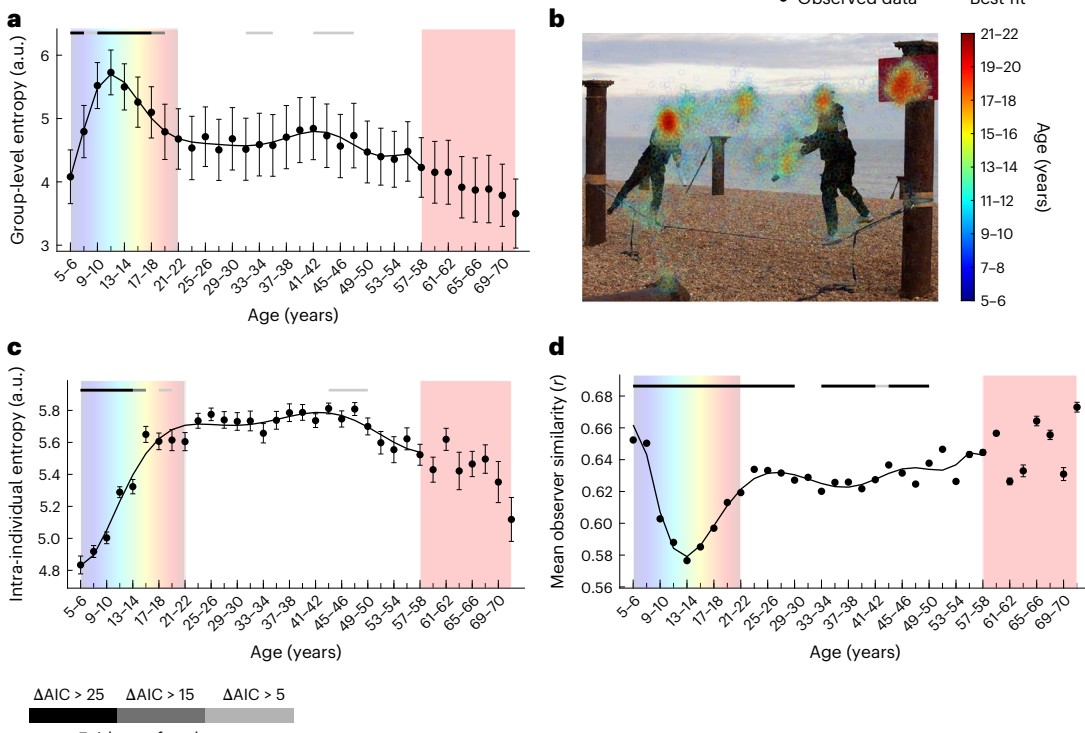

**Fig. 6 | Protracted development of inter-observer similarity. a**, The average entropy of group-level fixation maps across age bins. **b**, Fixation data from observers aged 5–22 overlaid as coloured circles on one example image. The colours correspond to age groups as indicated by the colour bar. Note the lower dispersion of hotter colours, indicating less dispersion at the group level for adult participants. **c**, The average entropy of observer-specific fixation maps across age bins. **d**, The mean observer similarity for each age bin, as indicated by the average pairwise correlation of observer-specific dwell time distributions across objects. A total of 6,720 participants were included in the analysis; for a more detailed distribution of participants across age groups, please refer to Fig. 2. Note that for **a**, **c** and **d**, each age bin spans two years. Due to space constraints, only every second tick mark is labelled. The red shading marks age bins with lower sensitivity to developmental changes, which were disregarded for curve fitting ($n < 90$ per bin). All error bars indicate ±1 s.e.m., and the lines indicate the best-fitting polynomial. The lines above each plot indicate the level of evidence for a given bin to be part of a broad linear change via shades of grey, as shown in the key (sliding window analysis; see main text and Methods). Panel **b** adapted from ref. 36 under a Creative Commons license CC BY 4.0.

## Protracted development of inter-observer similarity

Earlier work showed lower variability in gaze towards movie clips among adults than among infants[27] and children[28]. Here we found that intra-individual fixation dispersion increased until at least age 19–20. Initially, this was accompanied by an increase of individual differences, in terms of both object-directed dwell time distributions and group-level entropy. However, from age 13–14 this trend reversed, and inter-individual similarity steeply increased. Visual exploration became increasingly canonical, despite ongoing high levels of intra-individual fixation dispersion. This shows that visual exploration is individual in adolescents but becomes comparatively canonical in adults. As noted above, we speculate that this more canonical nature of adult gaze may reflect a slow accumulation of expertise for everyday scenes and increasingly similar visual experience.

## Potential protracted development of scene priors

A common thread across our findings is the potential role of visual experience and understanding. We speculate that the decreasing salience of touched objects may be driven by the developing ability to grasp the global meaning of (inter)actions from scene gist, without the need to foveate hands. The protracted increase of text salience is probably linked to education and an increasing share of text elements in the foveal diet. The horizontal bias may unfold along with spatial priors about typical object arrangements in everyday scenes. Most importantly, the unfolding canonical nature of scene exploration may point to the development of common templates for everyday scenes and their most relevant or informative regions. Individual visual

diets may become less idiosyncratic from the onset of out-of-home care[63]. Future studies could use mobile eye-tracking to test whether the intra-individual variance of visual diets increases with age, while the inter-individual variance decreases. Future studies may further compare the cultural specificity of canonical viewing behaviour at the group level. If culturally specific priors shape scene viewing, the decreasing variance among adults with a similar cultural background should be contrasted by increasing inter-cultural differences when comparing groups with different backgrounds.

We observed protracted development for semantic dwell-time salience, as well as for (distinct) semantic biases of the immediate saccade after image onset, and low-level oculomotor features such as the horizontal bias. Therefore, both bottom-up and top-down aspects of processing are probably subject to protracted development[35,38,40] and may be shaped by a self-reinforcing interplay between unfolding scene priors and active vision. Future research may use different tasks and neuroimaging to disentangle the role of bottom-up and top-down processing in the development of scene viewing.

## Limitations

The uneven age distribution of our sample resulted in limited sensitivity for developmental changes beyond age 56. At the same time, some variables (such as exploration tendency) appeared to develop across the entire lifespan. Also, we only included children from age five, but previous research indicates that earlier periods represent a time of remarkable plasticity and substantial cognitive and neural changes[64]. To enhance the sample size per age group and the robustness of related

point estimates, all fitted age trajectories and curve fits are based on age bins spanning two years. Future research may conduct experiments at an even larger scale or focus on smaller age ranges to trace developmental trajectories at a higher resolution (for example, age in months).

Furthermore, we estimated semantic salience biases and visuospatial biases using the OSIE 40 free-viewing task, which includes 40 natural scenes presented in a serial order. This method may lack generalizability and could introduce order effects. To mitigate these risks, we validated the current stimulus battery (OSIE 40) in a previous, well-powered and preregistered study[14]. This prior experiment explicitly tested the generalizability of individual gaze parameters for the OSIE 40 stimulus battery against a much larger set of scenes and confirmed the robustness of semantic gaze biases against stimulus order effects and their generalizability to out-of-sample scenes. We reanalysed these data regarding all metrics used in the current study and confirmed their robustness against effects of scene selection (Supplementary Information and Supplementary Fig. 8e). To further validate the robustness of our findings, we compared the size of developmental changes for each trajectory against the retest errors observed for corresponding means in independently acquired adult data. Observers in these studies took part in two free-viewing sessions using the same stimuli used in our main experiment. The image order was either fixed or shuffled across sessions. The results of these experiments showed retest errors that were orders of magnitude smaller than the observed developmental effects in the main sample, regardless of whether the stimulus order was fixed or shuffled between sessions (Supplementary Information pp. 21–22 and Supplementary Fig. 9).

The observed developmental trajectories are consistent across independent subsets of scenes (Supplementary Information and Supplementary Fig. 8b,c). Nevertheless, our results may not readily generalize to scenes or participants from an entirely different cultural background. Museum visitors from ages 5 to 72 are a more diverse group of participants than typical samples of university students but still far from representative of humanity. Both our stimuli and participants are probably marked by western, educated, industrialized, rich and democratic[65,66] biases. Future studies could cross stimuli and participants from different cultural backgrounds, which would be a strong test case for the hypothesis that developing gaze biases are linked to experience-dependent scene priors (see above).

Finally, our study employed a cross-sectional approach to examine developmental effects, which risks conflating age-related differences with cohort effects and other demographic changes not accounted for in our data collection. Future long-term longitudinal studies could trace individual developmental trajectories over time.

## Conclusion

Taken together, the data of thousands of individuals from ages 5 to 72 show that sematic salience, visual exploration and spatial saccadic biases unfold over the first two decades of life. The data further suggest that a crucial result of this protracted development is more canonical viewing behaviour across observers. Many of the underlying biological and environmental causes are unclear. Studying them appears key for understanding the general mechanisms of gaze.

## Methods

### Participants

From February 2022 to November 2023, a total of 13,984 datasets were collected at the German science museum Mathematikum, Giessen. Of these, 12,024 stemmed from individual participants who indicated no prior participation, and 10,614 of these proceeded to the free-viewing task (Fig. 1b). We excluded 6.49% of these due to incomplete data for more than half of the 40 presented images and a further 31.28% because their average calibration error was above one degree visual angle. This resulted in a final sample of 6,820 participants, of which 6,720 participants were 5 to 72 years of age (mean, 25.61; s.d. = 16.81; Fig. 2).

55.87% of this included sample identified as male, 40.84% as female and 3.3% as having diverse gender identities. The study was approved by the Justus Liebig Universität Fb06 Local Ethics Committee (lokale Ethik-Kommission des Fachbereichs 06 der Justus Liebig Universität Giessen; Ethics Protocol number LEK FB06 2021-0019), and the participants provided informed consent via a touchscreen. For minors, the procedure requested consent from a legal representative.

### Apparatus

The free-viewing task was programmed with Psychopy version 2020.1.2 (ref. 67) in Python version 3.6.10 (ref. 68). The stimuli were displayed on a Iiyama ProLite T1931SR-B5 touchscreen with a 1,280 × 1,024 pixel screen resolution and at a size of 1,000 × 750 pixels, which approximately corresponds to a visual angle of 27.5 × 20.6 degrees (dva). Eye movements were recorded from both eyes using a head-free Tobii 4c Eye Tracker (Tobii AB) operating in remote mode at a distance of 50–90 cm and with a sampling rate of 90 Hz. Previous research assessing the predecessor model, Tobii EyeX, reported an accuracy of <0.6 dva and a precision of <0.25 dva. Practice trials confirmed comparable performance for the 4c Eye Tracker, and the median validation accuracy for the present study was 0.6 dva (s.d. = 2.88).

### Procedure

Mathematikum visitors took part in our study from February 2022 to November 2023. The software interface was designed to enable participants to navigate the menu without external assistance. Figure 1a displays an image of the set-up; Fig. 1b displays the order of events during the experimental procedure. For a more detailed description, please refer to Supplementary Fig. 1.

The participants seated themselves in front of a touchscreen and initiated the process by selecting their preferred language (German or English). Subsequently, they provided informed consent, indicated their age (as integers), gender, prior participation (yes or no) and whether they wore glasses (yes or no). Following this, the participants underwent a five-point calibration and validation procedure. If the average validation error was above 0.7 dva, the procedure was repeated once more. To ensure correct positioning, a customized position guide interfaced with the eye-tracker and provided real-time feedback on whether the participant was within the trackable range. On average, participants sat at a distance of 61 cm from the screen. All viewing angles for event detections and error margins for area-of-interest (AOI) analyses (see below) were computed on the basis of the individual average distance.

Subsequently, the participants viewed an instructional video in which they were welcomed to the exhibit and instructed to 'look at the images in any way they want'. Then, 40 images were displayed at the centre of the screen for 3 s each, with a fixation cross displayed centrally between trials. A trial could commence only if a participant's gaze did not deviate by more than 2 dva from the fixation cross for 1 s. The presentation order of images was constant for all participants. We encouraged compliance and ensured data quality via the user guide, calibration and validation phases, as well as instructions to remain as still as possible and minimize head movement throughout. This was emphasized visually in an attention phase just before the experiment (Supplementary Fig. 1) and repeated verbally in the instructional video.

### Stimuli

We used the OSIE 40 (ref. 14) dataset, which includes 40 everyday scenes and corresponding pixel masks for 364 objects with binary labels for 12 semantic object features. We also used pixel masks for persons and inner-person features such as hands and limbs recently published in ref. 69. Individual gaze tendencies for several semantic features can be estimated reliably with this stimulus set[14]. Additional control analyses demonstrated that all other key variables examined in this study can be reliably estimated using the OSIE 40 (Supplementary Information and

Supplementary Fig. 8e). Finally, please note that the selected images do not depict any highly affective content; the strongest emotions conveyed are smiling faces.

## Data analyses

All preprocessing steps and statistical analyses were performed using MATLAB version R2019B (MathWorks). Raw eye-tracking data were converted to fixation and saccadic events using the event detection algorithm NH2010 by Holmqvist et al.[70]. The NH2010 was specified to identify fixations using a threshold on minimum fixation duration of 30 ms and a maximum velocity criterion of $80°\ s^{-1}$ as well as a minimum saccade duration of 20 ms (please refer to Supplementary Fig. 3c for example trace plots of (raw) eye-tracking data). For a given trial, we then removed fixations below 100 ms onset time to exclude intertrial fixations and saccades initiated before trial onset. We further removed all fixations with a duration <100 ms.

## Statistical approach

To examine the developmental trajectory of a given measure across age groups, we calculated individual averages and then averaged them within two-year age bins. This was done to increase the sample size of age bins and the robustness of related point estimates. We then computed fitted polynomials from zero to ten degrees to each curve. Among these candidate models, we computed the AIC differences as $\Delta i = AIC_i - AIC_{min}$, where $AIC_i$ stands for the AIC of the $i$th model, and $AIC_{min}$ represents the minimum AIC among all models. Subsequently, we selected the least complex model with $\Delta i < 4$ as the best-fitting choice. Due to the somewhat lower sample sizes for ages >56, we excluded these age groups from all curve-fitting analyses. We additionally confirmed the robustness of fits using cross-validation across independent subsamples of participants and images (Supplementary Information and Supplementary Figs. 8 and 9). The fitted parameters may inform future modelling studies[71,72] and aid age-tailored approaches[51]. To pinpoint periods of developmental change more specifically, we additionally used a sliding window approach. Here we computed the AIC difference ($\Delta AIC$) between first- and zero-degree polynomials (that is, the evidence for linear change over a constant) for windows of five consecutive age bins. We projected the evidence for change onto the two leftmost bins within a given window. This way, we evaluated the evidence for continuous linear change and interpreted its protracted nature in the most conservative fashion (evidence for change in each window is interpreted as change across at least the youngest age bins within the window). To cover the full age range, we projected the evidence for change onto all bins of the last window (that is, the window covering the oldest age range). To capture broader and less steep developmental changes, we report the results of the same analysis using a window size of ten consecutive age bins in the Supplementary Information; see Supplementary Fig. 7.

**Semantic salience.** To quantify semantic salience, we used two metrics of gaze towards objects with a given semantic feature: (1) the proportion of dwell time and (2) the proportion of first fixations after image onset among all fixations landing on labelled objects[13]. These were computed for each individual and then averaged for each age bin.

A tolerance margin was applied when assigning semantic object labels, allowing each fixation to receive labels for all objects within a radius of approximately 1 dva around a given fixation. This margin was applied to consider both the physiological extent of the fovea and possible eye-tracker inaccuracy. Fixations towards bodies were determined by removing fixations labelled as hand and head from person-labelled fixations.

**Oculomotor behaviour across development.** To examine oculomotor behaviour across age groups, we calculated the following metrics for each observer: (1) mean fixation eccentricity from the image centroid (in pixels); (2) visual exploration, indicated as the mean number of objects fixated; and (3) the mean fixation frequency in Hz. We also investigated the distribution of saccades across 60 directional bins and the proportion of saccades within ±45° of the horizontal meridian. Again, we averaged these metrics separately for each individual and age bin to trace developmental trajectories.

**Inter-observer similarity.** To test intra-individual fixation entropy, we computed the mean Shannon entropy of observer- and image-specific fixation maps and averaged the resulting values for each age bin. A given fixation map was created by applying a Gaussian filter with a standard deviation of 1 dva to each fixation. Similarly, we computed the group-level entropy on the basis of the Shannon entropy of joint fixation maps for all members of a given age group (and then averaged the entropy across images). Finally, to determine (dis)similarities in object-directed gaze, we first compiled the cumulative dwell time devoted to each labelled object across all scenes for every observer. We then computed all pairwise Pearson correlations between these individual gaze patterns in a given age bin and used the average of these correlations to determine typical pairwise similarity within the age bin.

## Data availability

The anonymized data are freely available via OSF at https://osf.io/ycrav.

## Code availability

The code to reproduce the presented findings is freely available via OSF at https://osf.io/ycrav.

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

## Acknowledgements

This work was supported by ERC Starting Grant No. 852885 INDIVISUAL. B.d.H. is further supported by Deutsche Forschungsgemeinschaft (German Research Foundation) Project No. 222641018–SFB/TRR 135 TP C9 and by 'The Adaptive Mind', funded by the Excellence Program of the Hessian Ministry of Higher Education, Science, Research and Art. We thank Xu et al. as well as Broda and de Haas for making their stimuli, pixel masks and metadata publicly available. We also thank the library staff of Justus-Liebig University for hosting the supplementary stimulus order experiment at short notice, as well as H. R. Wilkening, K. Schmidt, K. Caner, S. Fohs, T. Dalmis and Z. C. Demirkan for their assistance in collecting those data. Finally, we thank A. Beutelspacher, L. Samuel, and the workshop and entire team of the Mathematikum for a wonderful collaboration, as well as the participants in our study for their curiosity, time and data.

## Author contributions

M.L. and B.d.H. designed and implemented the experiment in collaboration with H.K., who assisted with Python coding. M.L. analysed the data and prepared the manuscript in collaboration with B.d.H., who administered and supervised the project. All authors approved the final article.

## Funding

## Competing interests

The authors declare no competing interests.

## Additional information

**Correspondence and requests for materials** should be addressed to Marcel Linka.

