## [Peer Review File · Nature Human Behaviour]

Protracted development of gaze behaviour

Corresponding Author: Dr Marcel Linka

Version 0:

Decision Letter:

3rd April 2024

Dear Mr Linka,

Thank you once again for your manuscript, entitled "Protracted development of gaze behaviour", and for your patience during the peer review process.

Your Article has now been evaluated by 3 referees. You will see from their comments copied below that, although they find your work of potential interest, they have raised quite substantial concerns. In light of these comments, we cannot accept the manuscript for publication, but would be interested in considering a revised version if you are willing and able to fully address reviewer and editorial concerns.

We hope you will find the referees' comments useful as you decide how to proceed. If you wish to submit a substantially revised manuscript, please bear in mind that we will be reluctant to approach the referees again in the absence of major revisions. We are committed to providing a fair and constructive peer-review process. Do not hesitate to contact us if there are specific requests from the reviewers that you believe are technically impossible or unlikely to yield a meaningful outcome.

To guide the scope of the revisions, the editors discuss the referee reports in detail within the team, including with the chief editor, with a view to (1) identifying key priorities that should be addressed in revision and (2) overruling referee requests that are deemed beyond the scope of the current study. We hope that you will find the prioritised set of referee points to be useful when revising your study. Please do not hesitate to get in touch if you would like to discuss these issues further.

1. Reviewer 2 mentions important concerns about the constant presentation order of the visual stimuli and the possibility of bias. To address this concern, please provide an additional validation eye-tracking experiment using new natural scenes and evaluate the invariance of the results on the used imagery dataset and the order of its presentation. Any new experiments should be preregistered and have at least 80% power to detect the smallest meaningful effect size (not observed effect sizes from the existing studies).

2. Our reviewers raise various concerns about the robustness of your findings. For example, Reviewer 1 is concerned that the dataset exhibits imbalances in age distribution, potentially leading to misinterpretations of trends, while Reviewer 2 asks that you evaluate the approximative functions on the validation set of participants. Please carefully address these concerns and provide additional robustness analyses.

3. Reviewer 1 and Reviewer 3 raise concerns regarding the polynomial curve fitting. Please clearly explain the necessity and efficacy of polynomial curve fitting for your data. In doing so, also address Reviewer 3's concerns that the fitted function might not be right.

4. While our reviewers are positive about the substantiveness of your data, they ask that you better emphasize the scientific advance of your work and embed it in the literature.

If you wish to submit a suitably revised manuscript, we would hope to receive it within 4 months. I would be grateful if you could contact us as soon as possible if you foresee difficulties with meeting this target resubmission date.

- Include a "Response to the editors and reviewers" document detailing, point-by-point, how you addressed each editor and referee comment. If no action was taken to address a point, you must provide a compelling argument. When formatting this document, please respond to each reviewer comment individually, including the full text of the reviewer comment verbatim followed by your

response to the individual point. This response will be used by the editors to evaluate your revision and sent back to the reviewers along with the revised manuscript.

- Highlight all changes made to your manuscript or provide us with a version that tracks changes.

Link Redacted

Thank you for the opportunity to review your work. Please do not hesitate to contact me if you have any questions or would like to discuss the required revisions further.

Sincerely,

[Redacted]

[Redacted] PhD

Nature Human Behaviour

Reviewer expertise:

Reviewer #1: polynomial curve fitting and eye-tracking

Reviewer #2: oculomotor biases and gaze preferences

Reviewer #3: oculomotor biases and gaze preferences ; eye-tracking and free viewing

REVIEWER COMMENTS:

Reviewer #1:

Remarks to the Author:

This paper presents an analysis of gaze behaviour development based on data collected from over 6500 subjects, representing a significant long-term effort. However, several issues warrant attention:

1. The dataset was collected automatically, raising concerns about data accuracy. For instance, some subjects may not follow instructions, resulting in noise in the data. Have any technical methods been employed to address this issue and ensure data quality?
2. There are inconsistencies in figure plotting. It is advisable for the author to maintain consistent y-axis ranges across related figures (e.g., in Figure 1A, 1B, 2A, and eta). In Figure 1A and 1B, the differing ranges hinder effective comparison across semantic categories.
3. The varying x-axis ranges in Figure 1A and 1B raise questions about the evidence for change.
4. Clarification is needed regarding the definition of age groups. Figure 1A presents age ranges of 5-6 and 9-10, but the range 6-9 is not addressed.
5. The dataset exhibits imbalances in age distribution, potentially leading to misinterpretations of trends. For instance, the low number of subjects in the 60-70 age range may introduce significant jitter in the data.
6. The rationale behind applying polynomial curve fitting is not clarified. With fewer than 20 age groups considered in the analysis, and trends already apparent in the dot plot, the necessity and efficacy of polynomial curve fitting should be justified. It is unclear how this fitting enhances understanding beyond what is already conveyed by the data.

Reviewer #2:

Remarks to the Author:

Key results:

The submitted manuscript deals with the gaze tendencies of a large sample of subjects (over 10,000) varying in age from 5 to 72 years with particular sensitivity in the range of 6 to 56 years over a limited set of semantic objects presented in a total of 40 natural scenes in an extensive eye-tracking research study. The authors justify the limited set of semantic objects and natural scenes chosen for the extensive eye-tracking research study by their previous publication (1). Authors aim to publicly provide the acquired dataset from their eye-tracking study which is undoubtedly considered as one of the considerable contributions of the given manuscript. The gaze measurements of the subjects were processed according to the state-of-the-art methods to classify fixations and saccades and derive a set of proposed metric results representing behavioural visual attention patterns of the subjects from a

wide age range, being unique among the state-of-the-art in the field. The presented results from such an extensive study are considered to be the most important contribution of the manuscript and are definitely of interest to the research community studying behavioural visual attention patterns and namely their relation to the age factor. The discussion of the results correctly confronts the results with the relevant state-of-the-art, providing unique research outcomes directly applicable to future research in the field.

(1) Linka, M. & de Haas, B. OSIEshort: A small stimulus set can reliably estimate individual 454 differences in semantic salience. *J Vis* 20, 13 (2020).

Validity and Originality:

To the best of my knowledge, the manuscript conclusions are original and the manuscript does not have flaws which should prohibit its publication.

The authors can find my comments and remarks regarding the manuscript divided into major and minor sections below. I would suggest the authors elaborate on their manuscript further to improve its quality for the final publication.

Major remarks:

- The abstract of the manuscript, as well as the introductory and contribution parts, describe the authors' contribution in a rather generalized manner. The main contribution of the manuscript is depicted in a more clear and concrete way in the later parts of the manuscript. I would like to suggest communicating the actual contribution to the state-of-the-art in the research field more explicitly in the abstract and the introductory part of the manuscript, as well.

- The manuscript is based on the eye-tracking research study with the imagery dataset originating in OSIE40 dataset containing natural scenes with affective higher-level visual stimuli with semantic labels. The elaboration of the authors on the methodology of selecting imagery data and containing affective stimuli would be beneficial for better understanding mainly in the context of the wide range of the participants' age distribution, thus resulting in differences in their top-down attention mechanisms.

- The authors provide interesting findings about canonical attention patterns related to socially affective natural visual stimuli observed in adulthood and the related development strategies converging towards these tendencies until late adolescence. Could the authors elaborate on the discussion regarding the convergence of visual attention tendencies towards canonical patterns in the context of bottom-up and top-down attention mechanisms and their interconnection with state-of-the-art knowledge about their development throughout the lifetime?

Ref: Li, L., Gratton, C., Fabiani, M., & Knight, R. T. (2013). Age-related frontoparietal changes during the control of bottom-up and top-down attention: an ERP study. *Neurobiology of aging*, 34(2), 477-488.

Amso, D., Haas, S., & Markant, J. (2014). An eye tracking investigation of developmental change in bottom-up attention orienting to faces in cluttered natural scenes. *PLoS one*, 9(1), e85701.

- The measured gaze data carry the measurement error produced by the eye-tracker, as stated by the authors in the manuscript. This error is subsequently manifested into computed fixations. Could the authors quantify and present this error for the given dataset and the discussed results? Furthermore, the measurement error may result in inaccurate fixation computations which may fall near the AOI borders. How did the authors take the measurement error into consideration when computing the intersections of fixations with the AOIs for further statistical analysis (error margins for AOI analyses mentioned in line 359)?

- There is evidence of gender differences in visual attention tendencies towards higher-level semantic objects in the state-of-the-art research in the field. Were these differences examined regarding the imbalanced gender representation in the data? Was there a potential bias observed to be pointed out in the manuscript?

Ref: Amon, M. J. (2015). Visual attention in mixed-gender groups. *Frontiers in psychology*, 5, 121908.

Gomez, P., von Gunten, A., & Danuser, B. (2019). Eye gaze behavior during affective picture viewing: Effects of motivational significance, gender, age, and repeated exposure. *Biological psychology*, 146, 107713.

- I highly appreciate the contribution of the manuscript regarding the approximation of the metric tendencies using polynomial developmental trajectories. The approximative function itself may serve as a mathematical model for the specific metric and a visual stimulus set, being a function of the participant's age. In order to support this contribution of the manuscript, I would suggest evaluating the approximative functions on the validation set of participants to prove the presented results and support the conclusions along with the quantified error declaration.

- The authors state that the presentation order of the visual stimuli was constant for all the participants. This could introduce a great bias in the measured gaze data and, thus, the presented and discussed results. I would like to question the authors whether the results are invariant on the limited set of visual images and order of image dataset presented to the participants. Could the authors provide an additional validation eye-tracking research study on a smaller participant sample using natural scenes which were unseen in the study currently presented in the manuscript and evaluate the invariance of the results on the used imagery dataset and the order of its presentation?

- The presented research study was designed with certain limitations, more specifically the number of natural scenes viewed by the participants, the specific order in which they were presented to the participants, and the various demographical and personal backgrounds of the participants resulting i.e. in biased top-down attention tendencies of the participants within and inter-group. The authors are encouraged to elaborate on the limitations of the design and measurements of their research study, as well as their impact on the results and conclusion statements presented in the manuscript. Moreover, I would encourage the authors to provide proof that their results and conclusions are invariant on the limitations of the design and the procedure of their research study and evaluation methodology.

Minor remarks:

- In the abstract of the manuscript, lines 28-29, the authors state that the development of scene viewing is surprisingly protracted. However, the development of visual attention patterns during childhood and late adolescence is of state-of-the-art knowledge. I would assume that the convergence of the visual attention patterns in the means of visual attractiveness of the socially affective stimuli towards the canonical state typical for adults is subject to protracted development discussed in the manuscript. Could the authors be more explicit about the facts in the manuscript?

Ref: Li, L., Gratton, C., Fabiani, M., & Knight, R. T. (2013). Age-related frontoparietal changes during the control of bottom-up and top-down attention: an ERP study. *Neurobiology of aging*, 34(2), 477-488.

Amso, D., Haas, S., & Markant, J. (2014). An eye tracking investigation of developmental change in bottom-up attention orienting to

faces in cluttered natural scenes. PloS one, 9(1), e85701.

- The introduction of the manuscript (lines 53-61) discusses visual attention tendencies regarding low-level and high-level features of natural scenes mainly from the point of view of bottom-up attention mechanisms. This should be explicitly stated in the manuscript. Moreover, I would suggest not neglecting the top-down attention mechanisms and their relation with the observed gaze behaviour, mainly the internal guidance originating in the internal state of the observer, memory, etc.

Ref: Theeuwes, J. (2010). Top-down and bottom-up control of visual selection. *Acta psychologica*, 135(2), 77-99.

Baluch, F., & Itti, L. (2011). Mechanisms of top-down attention. *Trends in neurosciences*, 34(4), 210-224.

- The authors suggest evaluating various metrics from 3-second long periods of image observation sequences to depict the gaze development phenomena in the manuscript. However, I would appreciate the authors explicitly stating the interconnection of the metrics proposal with the broader context of visual perception and visual attention strategies depicted in the manuscript in lines 62-87 and then in the Discussion section to point out the tested hypotheses early in the proposal.

- The polynomial developmental trajectory displayed in Figure 1/B does not reflect the absent evidence of change for ages over 56 and seems to wrongly suggest that the feature proportion increases for the elderly population. I assume this is the case also for Figure 1/A, Figure 2/B, Figure 3/B,C and Figure 4/A,D. I would suggest adjusting the polynomial developmental trajectory data visualization for the particularly non-sensitive age ranges (as stated in line 97).

- In line 353, the authors state that the information about previous participation, and wearing glasses was collected from the participants. Was the participant able to be involved in the study if previously participated? Was the participant able to be involved in the study if wearing glasses? If yes, could this affect the results?

- The authors validated the accuracy of the used eye-tracker and compared it to the predecessor eye-tracker version (lines 342-344). Could the authors provide information about the methodology of the validation practice trials executed?

- Does visualization in Figure 1 in the form of fixation map image overlay correctly display the eye-tracking measurement error quantified from the measured data? If yes, how the sigma of the Gaussian kernels for the plotted fixations was calculated for visualization purposes and also for fixation map smoothing to compute Shannon entropy (line 186)?

- In lines 352-353, the authors state the participants provided informed consent to participate in the study. How was this ensured regarding participants under 18 years for whom the consent should originate from their parent or legal representative?

- Could the authors give the reasoning for choosing namely the NH2010 fixation/saccadic detection algorithm among the others available?

Ref: Hooge I., et al. (2022). Fixation classification: how to merge and select fixation candidates. *Behavior Research Methods*. 54. 10.3758/s13428-021-01723-1.

Reviewer #3:

Remarks to the Author:

This study examines the development of gaze behavior on natural scenes. The study investigates several characteristics of gaze and compares them across age groups from early childhood to older age. It does so with an unusually large cohort of participants (6736). The study seems to be exploratory in its approach, as it examines a few different variables, without presenting a hypotheses or prediction to tie them together. I find a lot of value in this exploratory approach and in the large dataset (which I am happy to see, will be made publicly available). However, taking all the findings together, I am left unsure of what is the take-home message.

The main claim that the study is making is that gaze development is protracted, with some characteristics developing over the first two decades of life. I have a few concerns regarding this conclusion:

a. I think that for some of the measurements this conclusion derives from the polynomial curve fitting that was used. Looking at some of the figures, it seems as though the measurements reach a plateau earlier than the plateaus of the fitted function (which may suggest that the function is not the right fit). For example, in Figure 1A (touched and faces), in Figure 2B and in Figure 3 A+B, the data seems to plateau at early teens. While the fitted function suggests later. In figure 2B it can be clearly seen how the fitted function "forces" a developmental trend from late teens to early 30s. The collected data is higher than the fitted function for the early teens and then becomes lower for the 20s and until early 30s. This may create the illusion of a slow development, when in fact the plateau has been reached earlier than it seems (from around age 15, there doesn't seem to be any increase).

b. The introduction and the discussion do not tell a full story on what was previously known and what is currently revealed. I understand that the conclusion is that development is slow and lasts for two decades, but it remains unclear if this contradicts the current consensus. The findings are described one after the other, lacking a coherent theory or story to explain them and lacking the background to reveal whether (and if yes, why) they are groundbreaking.

In addition I have a few minor comments:

a. I might have missed it, but I can't find a summary of participants numbers in the different age groups, or their distribution across the age group. This is very important, and I think would be best of it comes as a figure. It's also important to know what the modulation of variability across the groups are.

b. Page 16, line 207 missing "and"

c. The number of participants is not reported consistently. It appears in the abstract as >6500. Then in the introduction it says "over 10,000" and it fact it was 6736

d. The sentence "fundamental aspects of adult gaze take decades of continuous development" from the abstract is focusing. It sounds like development "takes decades" when it fact it is around two decades at the most. So this may be misleading.

Version 1:

Decision Letter:

7th October 2024

Dear Mr Linka,

Thank you once again for your revised manuscript, entitled "Protracted development of gaze behaviour," and for your patience during the re-review process.

Your manuscript has now been evaluated by the same reviewers who evaluated your original manuscript. All reviewer feedback is included at the end of this letter. Although the reviewers found your manuscript to have improved during revision, they also raise some important outstanding concerns. We remain interested in the possibility of publishing your study in Nature Human Behaviour, but would like to consider your response to these outstanding concerns in the form of a revised manuscript before we make a decision on publication.

Specifically, we ask that you appropriately engage with the concerns raised by Reviewer 1 and address their concerns about the appropriateness of the curve fitting approach.

In sum, we invite you to revise your manuscript taking into account all reviewer and editor comments. We are committed to providing a fair and constructive peer-review process. Do not hesitate to contact us if there are specific requests from the reviewers that you believe are technically impossible or unlikely to yield a meaningful outcome.

We hope to receive your revised manuscript within 4-8 weeks. I would be grateful if you could contact us as soon as possible if you foresee difficulties with meeting this target resubmission date.

- Include a "Response to the editors and reviewers" document detailing, point-by-point, how you addressed each editor and referee comment. If no action was taken to address a point, you must provide a compelling argument. This response will be used by the editors and reviewers to evaluate your revision.
- Highlight all changes made to your manuscript or provide us with a version that tracks changes.

Link Redacted

We look forward to seeing the revised manuscript and thank you for the opportunity to review your work. Please do not hesitate to contact me if you have any questions or would like to discuss these revisions further.

Sincerely,

[Redacted]

[Redacted] PhD

[Redacted]
Nature Human Behaviour

Reviewer expertise:

Reviewer #1: polynomial curve fitting and eye-tracking

Reviewer #2: (development of) oculomotor biases and gaze preferences ; eye-tracking ; free viewing

Reviewer #3: (development of) oculomotor biases and gaze preferences ; eye-tracking ; free viewing

REVIEWER COMMENTS:

Reviewer #1 (Remarks to the Author):

Thank you for your response. While it has addressed some of my concerns, I remain uncertain about the approach to curve fitting.

I have carefully read the response and the manuscript, the curve fitting was applied to several metrics with age as the variable. However, the author does not specify whether the input values for age were integers or decimals. I assume they were integers (e.g.,

7 years old rather than 7.5). If that assumption is correct, I question the necessity of polynomial fitting, which seems to serve only to enhance the visualization. For instance, if we want to estimate the dwell time for 8-year-olds, using the average value for all individuals aged 8 would likely be more reliable than relying on a value derived from the curve fitting. Moreover, the absence of decimal data reduces the model's confidence in estimating values for intermediate ages, such as 7.5 years.

I also checked the author's discussion of the R2 question about Figure S7, where cross-validation was conducted by splitting the dataset in half. The results of this experiment is unreliable, as dividing the data into two equal parts may still retain factors such as the presentation order of stimuli. This similarity between subsets naturally leads to comparable results.

More importantly, I find it problematic that the paper's analyses focus exclusively on age as the primary variable. As other reviewers have pointed out, the order of stimulus presentation seems to have a noticeable effect, as shown in Figure S8. Yet, this issue was not considered in the design of the author's other experiments, which seems inappropriate.

This is why, in my initial review, I suggested that the author provide a clear justification for the use of curve fitting. Given that the experimental setup is fixed, e.g. fixed presentation order, the fitted curve only reflects variation within the fixed conditions, rather than offering a generalized result. Furthermore, when the age data consists solely of integers, there seems to be even less need for fitting, as the average values for each age group can be used directly.

Minor: I observed in Figure S2 that there are significant deviation in gaze accuracy error across different age groups, with notable variability for children aged 5 to 6. I wonder whether this could introduce bias into the analysis of younger children.

Reviewer #2 (Remarks to the Author):

I would like to express my thanks for the authors response to the reviewers' comments and my satisfaction with the provided thorough evaluation and manuscript modifications regarding the addressed concerns. To improve the manuscript for publication, I would add up the following minor comments:

- page 2 line 28: The authors state previous evidence points out 'mature' viewing behaviour from about 8 years of age. However, I find the word 'mature' in the context misleading and uncommon in the literature in connection with the mentioned age. Could the authors provide reference on common usage of this term in the literature, or alter the statement to be conform with the consensus?
- pages 4-5 lines 96,111: The authors use two terms to reference the target sample of their research: 'adolescence' and 'teenage'. While these are generally understandable to the audience, I would recommend to unify the terms in the manuscript, unless the authors introduce both these terms intentionally and refer to various target samples.
- page 8 lines 147-149: "Overall, the developmental trajectory of dwell time proportions was best fit by a 3rd degree polynomial, while that of first fixations was best fit by a linear model with a slight positive slope." - I would recommend the authors reference the methodology to find the best fit here, and in the appropriate occurrences of another such assumptions in the manuscript. Therefore, the reader has all the information needed to understand the authors' claims always directly in the place.
- I highly appreciate introducing results regarding the protracted development of gender differences and I find the results and corresponding discussion interesting. I would recommend that the manuscript points out the state-of-the-art in this particular domain earlier in the introduction and incorporates the proposed evaluation objectives into the story of the paper contribution in the introductory parts because the interconnection of the provided evaluation with the aims of the manuscript is now missing.
- Figure 1C: bottom-left graph contains blue-red marks without legend; the evidence of change in the visualizations is hardly visible except the strongest evidence, moreover gray patches conflict with the colour scale used for the evidence of change
- Figure 2B, 3A-C: This Figure should contain more sophisticated visual attributes mapping, so the trendline can be visually better comparable to the observed trends (i.e. by using colour attributes).
- I would recommend the authors to thoroughly proof-read the paper before submission for the publication (see inconsistency in spelling of the word 'behavior' in page 19, line 353, etc.)

Reviewer #2 (Remarks on code availability):

I would encourage the authors to provide the raw eyetracking data publicly available alongside with the publication of the manuscript to elevate the contribution of their research (future raw data publication promise in readme file).

Reviewer #3 (Remarks to the Author):

The authors have made considerable effort to improve the manuscript. I was satisfied with the answers to my questions and the resulting revisions. I find the paper ready for publication.

Version 2:

Decision Letter:

Our ref: NATHUMBEHAV-23124326B

28th January 2025

Dear Dr. Linka,

Thank you for submitting your revised manuscript "Protracted development of gaze behaviour" (NATHUMBEHAV-23124326B). It has now been seen by the original referees and their comments are below. As you can see, the reviewers find that the paper has improved in revision. We will therefore be happy in principle to publish it in Nature Human Behaviour, pending minor revisions to satisfy the referees' final requests and to comply with our editorial and formatting guidelines.

We are now performing detailed checks on your paper and will send you a checklist detailing our editorial and formatting requirements within two weeks. Please do not upload the final materials and make any revisions until you receive this additional information from us.

Sincerely,

[REDACTED]

[REDACTED] PhD

Nature Human Behaviour

Reviewer #1 (Remarks to the Author):

The author has made considerable efforts to improve the manuscript in this submission. I appreciate the work put into the new figure plots and the detailed experimental analysis. The manuscript now appears ready for publication.

Version 3:

Decision Letter:

Dear Dr Linka,

We are pleased to inform you that your Article "Protracted development of gaze behaviour", has now been accepted for publication in Nature Human Behaviour.

We welcome the submission of potential cover material (including a short caption of around 40 words) related to your manuscript; suggestions should be sent to Nature Human Behaviour as electronic files (the image should be 300 dpi at 210 x 297 mm in either TIFF or JPEG format). Please note that such pictures should be selected more for their aesthetic appeal than for their scientific content, and that colour images work better than black and white or grayscale images. Please do not try to design a cover with the Nature Human Behaviour logo etc., and please do not submit composites of images related to your work. I am sure you will

understand that we cannot make any promise as to whether any of your suggestions might be selected for the cover of the journal.

With best regards,

[Redacted]

[Redacted] PhD

[Redacted]
Nature Human Behaviour

P.S. Click on the following link if you would like to recommend Nature Human Behaviour to your librarian
<http://www.nature.com/subscriptions/recommend.html#forms>

** Visit the Springer Nature Editorial and Publishing website at http://editorial-jobs.springernature.com?utm_source=ejp_NHumB_email&utm_medium=ejp_NHumB_email&utm_campaign=ejp_NHumB for more information about our career opportunities. If you have any questions please click [here](mailto:editorial.publishing.jobs@springernature.com).

Dear Dr. [REDACTED]

Thank you very much for the opportunity to revise our manuscript titled "Protracted development of gaze behaviour."

Please find attached our point-by-point response to the key priorities raised by the editorial team, as well as all points raised by the reviewers. All changes made to the manuscript and Supplementary Materials are highlighted in yellow throughout the document. Should any questions or concerns arise, please do not hesitate to contact us.

1. Reviewer 2 mentions important concerns about the constant presentation order of the visual stimuli and the possibility of bias. To address this concern, please provide an additional validation eye-tracking experiment using new natural scenes and evaluate the invariancy of the results on the used imagery dataset and the order of its presentation. Any new experiments should be preregistered and have at least 80% power to detect the smallest meaningful effect size (not observed effect sizes from the existing studies).

We appreciate the concern regarding potential stimulus and presentation order biases

To address this concern, we have validated the generalizability of the current stimulus battery (OSIE 40) in a previous, well-powered ($n > 100$), and preregistered study. This experiment tested the generalizability of individual gaze parameters for the stimulus battery used in the current study against a much larger set of scenes. It also tested the consistency of estimated individual gaze parameters for these stimuli across different stimulus orders. We have now reanalyzed this data concerning all metrics used in the current study and confirmed their robustness against effects of stimulus order as well as their generalizability to out-of-sample scenes (see revised Supplementary Figures and text).

Additionally, we directly examined the robustness of developmental trajectories in the existing dataset. We analysed the developmental trajectories for odd and even scenes, confirming that all our conclusions generalized between independent subsets of scenes (see attached figures and results).

2. Our reviewers raise various concerns about the robustness of your findings. For example, Reviewer 1 is concerned that the dataset exhibits imbalances in age distribution, potentially leading to misinterpretations of trends, while Reviewer 2 asks that you evaluate the approximative functions on the validation set of participants. Please carefully address these concerns and provide additional robustness analyses.

To improve the robustness of fits, we now followed the reviewer suggestion and excluded age bins over 56 for the curve-fitting (and highlighted the lack of power for these bins in all figures). This adjustment has resulted in more accurate fits, including a correction of the inconsistencies with raw data noted by Reviewer 1.

Moreover, we have tested the robustness of our curve-fitting approach by performing split-half cross-validations for 12 key metrics included in the study. For each metric, we computed correlations between the fitted age trajectories for one half of the participants and the observed bin-wise means of the other half. All cross-validated correlations were close to estimated noise ceilings,

underscoring the robustness and validity of the curve-fitting approach (new Figure S7).

3. Reviewer 1 and Reviewer 3 raise concerns regarding the polynomial curve fitting. Please clearly explain the necessity and efficacy of polynomial curve fitting for your data. In doing so, also address Reviewer 3's concerns that the fitted function might not be right.

We acknowledge that some of our curve fits may have been biased due to the lower sensitivity of age bins greater than age 56. As a result, we have excluded these age groups from our curve fitting analyses, leading to improved fits across age trajectories (see above). Additionally, we have now motivated the use of polynomial curve fitting more clearly in the methods section.

Finally, based on Reviewer 3's feedback, we revised our sliding-window approach to identify significant phases of change for a given age trajectory. Our initial approach resulted in smoothing, at the risk of concluding overly protracted development. We have now adopted a highly conservative approach, evaluating the evidence for change between any two consecutive age bins as the evidence of linear change in the window starting with these two bins (total window size still 5 bins). This approach is maximally conservative and the resulting evidence can be interpreted as change *at least* up until the indicated age of development. The resulting significance patterns better align with visual data inspection and shifted some developmental inflection points back to early adulthood or teenage. We updated all relevant sections of the manuscript accordingly

4. While our reviewers are positive about the substantiveness of your data, they ask that you better emphasize the scientific advance of your work and embed it in the literature.

We have revised the abstract, introduction, and discussion to better embed our work within the literature and highlight our unique contribution.

Reviewer #1:

Remarks to the Author:

This paper presents an analysis of gaze behaviour development based on data collected from over 6500 subjects, representing a significant long-term effort. However, several issues warrant attention:

1. The dataset was collected automatically, raising concerns about data accuracy. For instance, some subjects may not follow instructions, resulting in noise in the data. Have any technical methods been employed to address this issue and ensure data quality?

We thank the reviewer for this question. We have implemented several methods to ensure data quality. First, we designed the experimental procedure to encourage instruction compliance. Specifically, we implemented a user guide phase that participants could only pass if they were in the correct position (i.e. at the right distance from the tracker and within the Tobii tracker box; see Supplementary Figure S1). Additionally, we instructed participants to 'sit still and move as little as possible' via a video instruction and repeated 'attention sign' reminders.

Further, participants had to complete a calibration and validation routine, which was repeated if the first validation showed a calibration error > 0.7 degrees visual angle (dva). During the free viewing task, participants could only proceed to the next trial if they fixated centrally for 1 second, with a tolerance margin of 2 degrees. If they did not do so for more than 10s, they were reminded to look at the central cross and the session was aborted if they did not do so. Thus, only participants showing a sufficient degree of compliance and data accuracy were able to complete a session. We included the entire procedure in Figure S1 (Supplementary Procedure) and describe it in detail. We now additionally refer to this more explicitly in the main text on page 24, l. 446-449.

We encouraged data quality and compliance via the user guide, calibration and validation phases, as well instructions to remain as still as possible and minimize head movement throughout. This was emphasized visually in an attention phase just before the experiment (see Figure S1) and repeated verbally in the instructional video.

Most importantly, for our analyses, we only included participants with an average validation accuracy of < 1 dva deviation from the nominal inter-trial fixation target (median deviation across all included participants = 0.48 dva). Further, we excluded participants with data for less than half of the images. To make this clearer, we have now visualized the overall distribution of mean validation accuracy as well as the median validation accuracy per age bin (Figure S2) and describe the validation accuracy data in more detail in the Supplementary Materials.

Validation accuracy

Figure S2 illustrates the median validation accuracy per age group for the included (green) and excluded (red) data in our study. The entire collected sample of 13,984 datasets has a median validation accuracy of 0.6 dva. For all analyses in this study, we used an individual threshold of < 1 dva as the inclusion criterion for further processing. Using this criterion, the remaining datasets show a median accuracy of 0.48 dva. When computing the validation accuracy per age group (bin), we observe weaker validation accuracies around the age groups 5-6 to 13-14 (median = 0.64 dva) for all collected datasets. Importantly, when only analysing included participants with a mean validation accuracy < 1 dva, this difference disappears (median = 0.5 dva)

Figure S2. Eye-tracking validation accuracy across age. Panel A shows a bar plot displaying the median accuracy in degrees of visual angle (dva) for each age bin. Values in green represent values computed including participants with an accuracy of less than 1 dva, whose data were included in all analyses of this study. Values in red represent median scores that include data from participants with a mean accuracy score greater than 1 dva. Panel B displays the individual mean validation accuracy scores in dva across age. Each scatter point represents one individual. Green scatter points indicate participants included in all analyses, with a validation accuracy of less than 1 dva, highlighted in the zoomed-in scatter plot on the upper right. Red scatter points represent participants with a mean validation score greater than 1 dva.

2. There are inconsistencies in figure plotting. It is advisable for the author to maintain consistent y-axis ranges across related figures (e.g., in Figure 1A, 1B, 2A, and eta). In Figure 1A and 1B, the differing ranges hinder effective comparison across semantic categories.

We understand that the varying scales may hinder effective comparisons across semantic categories. To address this, we have now added panels to Figure 1 (see Panel A and B below point 3) that include trajectories for all tested semantic categories on the same scale. We opted to additionally keep panels ‘zooming’ in on a given semantic category, which more effectively communicate relative change across age, which is of particular interest in the context of our study.

3. The varying x-axis ranges in Figure 1A and 1B raise questions about the evidence for change.

We suppose this comment refers to y-axis ranges as well (x-axes indicating age are identical). We fully concede that the amplitude of absolute changes varies across semantic categories and now communicate this more clearly (see above). Crucially, the evidence for change across age is strong for all developing categories (i.e. all but bodies). This is true in statistical terms, as indicated by the AIC values in favour of change and by the magnitude of changes in relation to the much smaller uncertainty about bin-wise means (error shades of curves in Figure 1). It is also true when considering the magnitude of changes relative to the baseline of 5-6 year olds. They are on the order of 10% or more for all

developing categories, and can be considered extreme for text, with a tripling of dwell time until adulthood. In short, the evidence for protracted development is highly significant, both in statistical and practical terms.

Figure 1. Protracted development of semantic salience. Line plots display the mean proportion of dwell time and first fixations directed towards objects of four semantic features across age bins displayed on one axis (A-B) and zoomed in on each semantic category separately (C-D). Please note that each age bin spans two years. Due to space constraints, only every second tick is labelled, and these are marked in bold for easier identification. Semantic categories are indicated by colours, as shown in the inset. Error shades indicate ± 1 standard error of the mean (SEM). Dotted lines show the corresponding best fitting polynomials. The lines above the x-axes indicate the level of evidence for a given bin to be part of broad linear change via shades of grey, as shown in the inset (sliding window analysis, cf. main text and methods). Gray patches mark age bins with lower sensitivity to developmental changes, which were disregarded for curve fitting ($n < 90$ per bin). The left-hand side of panel (E) displays object pixel masks for an example image, with colors corresponding to categories in A - D (cf. inset, objects outside these categories shown in light gray). The middle and right-hand images display alpha-blended fixation overlays for the same image, showing all fixations (middle) or only first fixations after image onset (right-hand side) for two age groups. Fixation overlays shown in blue and red correspond to ages 5-6 and 19-20, respectively. These age groups are marked correspondingly with blue and red rhombuses above the x-axes in Panels (C) and (D).

4. Clarification is needed regarding the definition of age groups. Figure 1A presents age ranges of 5-6 and 9-10, but the range 6-9 is not addressed.

We thank the reviewer for pointing this out and understand that our tick labeling was not the most intuitive. In all figures regarding age comparisons (including Figure 1A), each age bin pools two years of age. To aid legibility, only every other bin is labeled. For example, the age bin 5-6 includes ages 5 and 6, then the following unlabeled tick includes ages 7 and 8, and the next labeled tick shows data from participants aged 9 and 10. We now made this explicit and more intuitive by printing the labeled ticks bold and including a sentence in the figure captions to clarify the labelling (see Figure 1 above for an example).

5. The dataset exhibits imbalances in age distribution, potentially leading to misinterpretations of trends. For instance, the low number of subjects in the 60-70 age range may introduce significant jitter in the data.

Please note that the curve fitting analyses used the full dataset rather than aggregated averages and thus takes the uneven age distribution into account (bins with fewer participants will contribute less to the overall error metric). Furthermore, we have excellent precision for the estimation of averages in the age range 5-56 (with an $n > 90$ for each bin), which forms the basis of our conclusions. We agree that participant numbers and sensitivity for the older age range 56-72 are more limited. We have now excluded this age range for the curve fitting and mention this caveat on page 25, l. 473-475:

Due to the somewhat lower sample sizes for ages > 56 , we excluded these age groups from all curve-fitting analyses

Additionally, we mark the limited data coverage for ages 56-72 with shading in the figures and explain this in figure captions (see below for an example).

Figure 4. Protracted development of inter-observer similarity. Panel A shows the average entropy of group-level fixation maps across age bins. Panel B shows fixation data from observers aged 5-22 overlaid as colored circles on one example image. Colors correspond to age groups as indicated by the colorbar. Note the lower dispersion of hotter colors, indicating less dispersion on the group level for adult participants. Panel C shows the average entropy of observer-specific fixation maps across age bins. Panel D shows the mean observer similarity for each age bin, as indicated by the average pairwise correlation of observer-specific dwell-time distributions across objects. **Note that for panels A, C and D, each age bin spans two years. Due to space constraints, only every second tick is labelled, and these are marked in bold for easier identification. Gray patches mark age bins with lower sensitivity to developmental changes, which were disregarded for curve fitting ($n < 90$ per bin).** All error shades indicate ± 1 standard error of the mean (SEM) and the dotted lines the best fitting polynomial. The lines above the x-axes indicate the level of evidence for a given bin to be part of broad linear change via shades of grey, as shown in the inset (sliding window analysis, cf. main text and methods).

Finally, we now emphasize this point more in the limitations section on page 19, l. 365-367.

The uneven age distribution of our sample resulted in limited sensitivity for developmental changes beyond age 56. At the same time, some variables (such as exploration tendency) appeared to develop across the entire life span

6. The rationale behind applying polynomial curve fitting is not clarified. With fewer than 20 age groups considered in the analysis, and trends already apparent in the dot plot, the

necessity and efficacy of polynomial curve fitting should be justified. It is unclear how this fitting enhances understanding beyond what is already conveyed by the data.

The motivation for applying polynomial curve fitting is that the model parameters can serve as a model for the specific metric across different ages. This approach may be beneficial for work modeling and predicting natural human gaze behavior (Kümmerer et al., 2016, 2017). We have now elaborated on this in greater detail in the method section on page 25, l. 476-477.

The fitted parameters may inform future modelling studies (Kümmerer et al., 2016, 2017) and aid age-tailored approaches (Strauch et al., 2023).

Reviewer #2:

Remarks to the Author:

Key results:

The submitted manuscript deals with the gaze tendencies of a large sample of subjects (over 10,000) varying in age from 5 to 72 years with particular sensitivity in the range of 6 to 56 years over a limited set of semantic objects presented in a total of 40 natural scenes in an extensive eye-tracking research study. The authors justify the limited set of semantic objects and natural scenes chosen for the extensive eye-tracking research study by their previous publication (1). Authors aim to publicly provide the acquired dataset from their eye-tracking study which is undoubtedly considered as one of the considerable contributions of the given manuscript. The gaze measurements of the subjects were processed according to the state-of-the-art methods to classify fixations and saccades and derive a set of proposed metric results representing behavioural visual attention patterns of the subjects from a wide age range, being unique among the state-of-the-art in the field. The presented results from such an extensive study are considered to be the most important contribution of the manuscript and are definitely of interest to the research community studying behavioural visual attention patterns and namely their relation to the age factor. The discussion of the results correctly confronts the results with the relevant state-of-the-art, providing unique research outcomes directly applicable to future research in the field.

(1) Linka, M. & de Haas, B. OSIEshort: A small stimulus set can reliably estimate individual 454 differences in semantic salience. *J Vis* 20, 13 (2020).

Validity and Originality:

To the best of my knowledge, the manuscript conclusions are original and the manuscript does not have flaws which should prohibit its publication.

The authors can find my comments and remarks regarding the manuscript divided into major and minor sections below. I would suggest the authors elaborate on their manuscript further to improve its quality for the final publication.

We thank the reviewer and will address each point in detail below.

Major remarks:

- The abstract of the manuscript, as well as the introductory and contribution parts, describe the authors' contribution in a rather generalized manner. The main contribution of the manuscript is depicted in a more clear and concrete way in the later parts of the manuscript. I would like to suggest communicating the actual contribution to the state-of-the-art in the

research field more explicitly in the abstract and the introductory part of the manuscript, as well.

We thank the reviewer for pointing this out and fully agree that highlighting the explicit contribution is important. To better embed our research question within the literature and highlight our distinct contribution, we have revised parts of the abstract (pages 2, l. 27-29):

How does scene viewing develop? Previous evidence is limited and suggests viewing behaviour may be mature from about 8 years of age. Here, we present data from $n = 6730$ subjects from 5-72 years of age, freely viewing 40 natural scenes.

the introduction section (pages 3-4, l. 64-82).

A central question in visual perception is whether and how oculomotor biases and attentional preferences develop. Infants(Franchak et al., 2016) and children(Rider et al., 2018) watching movie clips exhibit greater interindividual variability than adults, suggesting that common attentional biases are learned. Although gaze preferences for faces and social information are present in infants(Andravizou et al., 2008; Gluckman & Johnson, 2013; Johnson et al., 1991, 2015; Kelly et al., 2019) and early childhood (Franchak et al., 2016; Frank et al., 2011), it remains poorly understood how they evolve from thereon. Amso et al., (2014) have reported the development of gaze towards faces to plateau around age 8 (Amso et al., 2014). Another study found lower text salience and higher salience of hands and objects being touched in preschoolers compared to adults(Linka et al., 2023) . More basic patterns of oculomotor behaviour also seem to develop during childhood. Saccade amplitudes are higher and fixation durations are shorter at later stages of childhood (age 4-8) compared to age 2 (Helo et al., 2014) and the tendency to fixate on central image regions is weaker at age 8 than 4 (Krishna et al., 2018). The horizontal bias seems to be present in infants, albeit likely to a lesser extent than in adults(Van Renswoude et al., 2016). While most studies report the development of basic oculomotor behavior in children, other research finds no significant differences in visual exploration and fixation frequency between children and young adults(Acik et al., 2010). Together, these studies suggest that basic visuospatial and semantic salience biases differ between young children and adults but may mature as early as age 8. Overall, however our understanding of gaze development is severely limited by a scarcity of data, with most studies comparing two or three age groups with limited sample sizes, which rarely cover adolescence and do not allow a continuous comparison of gaze across development. It thus remains unclear whether and how scene viewing changes beyond early childhood.

Page. 4, l. 95-97:

Here, we aimed to test whether scene viewing matures early or develops into adolescence and beyond. Therefore, we traced the developmental trajectories

of semantic salience and basic oculomotor biases continuously and across a wide age range.

Page 5, l. 109-113:

We find strong evidence for a protracted development of attentional preferences and basic oculomotor biases extending into teenage and young adulthood. Furthermore, children viewed scenes in more idiosyncratic ways, , which gradually gave way to more canonical patterns, which fully matured in the mid-20s.

And the discussion section (pages 14-15, l. 234-240):

Previous studies using small and focussed samples have reported differences in semantic salience (Amso et al., 2014; Franchak et al., 2016; Linka et al., 2023) and facets of oculomotor behaviour (Helo et al., 2014; Krishna et al., 2018) between adults and children younger than eight years of age. Here, we found surprisingly protracted development for the semantic salience of text, faces, and touched objects, as well as for visual exploration and basic oculomotor biases, such as the central and horizontal bias, all of which continuously developed into adolescence and for some measures up until young adulthood.

- The manuscript is based on the eye-tracking research study with the imagery dataset originating in OSIE40 dataset containing natural scenes with affective higher-level visual stimuli with semantic labels. The elaboration of the authors on the methodology of selecting imagery data and containing affective stimuli would be beneficial for better understanding mainly in the context of the wide range of the participants' age distribution, thus resulting in differences in their top-down attention mechanisms.

The OSIE40 stimuli set was derived from the larger OSIE 700 image dataset of complex scenes with extensive pixel masks and semantic annotations (Xu et al., 2014; de Haas et al., 2018). Crucially, the OSIE 40 was constructed and validated as an efficient test of individual gaze biases in a previous publication (Linka & de Haas, 2020). We now refer to our validation study in more detail in the methods section on page 24, l. 454-456:

Additional control analyses demonstrated that all other key variables examined in this study can be reliably estimated using the OSIE40, regardless of image order (see Supplementary Materials).

Moreover, the OSIE images are not affective in nature (compared to affective image databases like the IAPS). The OSIE40 images were explicitly selected to avoid strong affective connotations, with the strongest valence probably conveyed by smiling faces. We now clarify this in the methods section on page 24, lines 457-458.

Finally, please note that the selected images do not depict any highly affective content; the strongest emotions conveyed are smiling faces.

- The authors provide interesting findings about canonical attention patterns related to socially affective natural visual stimuli observed in adulthood and the related development strategies converging towards these tendencies until late adolescence. Could the authors elaborate on the discussion regarding the convergence of visual attention tendencies towards canonical patterns in the context of bottom-up and top-down attention mechanisms and their interconnection with state-of-the-art knowledge about their development throughout the lifetime?

Ref: Li, L., Gratton, C., Fabiani, M., & Knight, R. T. (2013). Age-related frontoparietal changes during the control of bottom-up and top-down attention: an ERP study.

Neurobiology of aging, 34(2), 477-488.

Amso, D., Haas, S., & Markant, J. (2014). An eye tracking investigation of developmental change in bottom-up attention orienting to faces in cluttered natural scenes. PloS one, 9(1), e85701.

We thank the reviewer for pointing us to these important papers. The findings in Amso et al. highlight an interesting potential explanation for some of our findings: the increasing developmental trajectory in the proportion of face-directed first fixations might be driven by enhanced bottom-up salience for faces during childhood in the first second of viewing time, as demonstrated by Amso et al. (2015). We have now discussed this and other implications of our findings regarding the development of top-down and bottom-up attentional mechanisms on page 19, l. 357-363.

Interestingly, we observed protracted development for semantic dwell-time salience, as well as for (distinct) semantic biases of the immediate saccade after image onset, and low-level oculomotor features like the horizontal bias. Therefore, both, bottom-up and top-down aspects of processing likely are subject to protracted development (Acik et al., 2010; Amso et al., 2014; Krishna et al., 2018) and may be shaped by a self-reinforcing interplay between unfolding scene priors and active vision. Future research may use different tasks and neuroimaging to disentangle the role of bottom-up and top-down processing in the development of scene viewing.

- The measured gaze data carry the measurement error produced by the eye-tracker, as stated by the authors in the manuscript. This error is subsequently manifested into computed fixations. Could the authors quantify and present this error for the given dataset and the discussed results? Furthermore, the measurement error may result in inaccurate fixation computations which may fall near the AOI borders. How did the authors take the measurement error into consideration when computing the intersections of fixations with the AOIs for further statistical analysis (error margins for AOI analyses mentioned in line 359)?

The median validation accuracy for included participants was 0.48 dva. We now describe this in greater detail in the Supplementary Materials:

Validation accuracy

Figure S2 illustrates the median validation accuracy per age group for the included (green) and excluded (red) data in our study. The entire collected

sample of 13,984 datasets has a median validation accuracy of 0.6 dva. For all analyses in this study, we used an individual threshold of <1 dva as the inclusion criterion for further processing. Using this criterion, the remaining datasets show a median accuracy of 0.48 dva. When computing the validation accuracy per age group (bin), we observe weaker validation accuracies around the age groups 5-6 to 13-14 (median = 0.64 dva) for all collected datasets. Importantly, when only analysing included participants with a mean validation accuracy <1 dva, this difference disappears (median = 0.5 dva).

Figure S2. Eye-tracking validation accuracy across age. Panel A shows a bar plot displaying the median accuracy in degrees of visual angle (dva) for each age bin. Values in green represent values computed including participants with an accuracy of less than 1 dva, whose data were included in all analyses of this study. Values in red represent median scores that include data from participants with a mean accuracy score greater than 1 dva. Panel B displays the individual mean validation accuracy scores in dva across age. Each scatter point represents one individual. Green scatter points indicate participants included in all analyses, with a validation accuracy of less than 1 dva, highlighted in the zoomed-in scatter plot on the upper right. Red scatter points represent participants with a mean validation score greater than 1 dva.

When assigning objects and semantic labels to fixations, we used an overlap with an error margin of 0.5 d.v.a. By applying this error margin, each fixation was assigned the labels of all objects falling within a radius of ~0.5 dva surrounding the pixel coordinates of the fixation. Please note the motivation behind this is two-fold: First to take into account the physiological extend of the fovea and the need to account for uncertainty due to measurement inaccuracies. Even in worst-case scenarios where the fovea was centered 0.5-1 dva from an object it can be legitimately considered to be foveated due to the extent of the fovea. This information has now been added to the methods section on page 26, lines 491-494.

A tolerance margin was applied when assigning semantic object labels, allowing each fixation to receive labels for all objects within a radius of approximately 0.5 dva around a given fixation. This margin was applied to consider both, the physiological extend of the fovea and possible eyetracker inaccuracy.

- There is evidence of gender differences in visual attention tendencies towards higher-level semantic objects in the state-of-the-art research in the field. Were these differences examined regarding the imbalanced gender representation in the data? Was there a potential bias observed to be pointed out in the manuscript?

Ref: Amon, M. J. (2015). Visual attention in mixed-gender groups. *Frontiers in psychology*, 5, 121908.

Gomez, P., von Gunten, A., & Danuser, B. (2019). Eye gaze behavior during affective picture viewing: Effects of motivational significance, gender, age, and repeated exposure. *Biological psychology*, 146, 107713.

We thank the reviewer for the suggestion and have now examined gender differences for (1) developmental trajectories of semantic salience and (2) semantic salience pooled across age groups.

Our findings show similar age trajectories for both male and female participants in terms of dwell time and first fixations towards objects across all included semantic categories. Within specific age groups, significant gender differences were observed. Surprisingly, and contrasting with earlier findings in adults reporting a visual preference for social stimuli in females (Lutchmaya & Baron-Cohen, 2002; Proverbio, 2017; Proverbio et al., 2008) males exhibited a larger proportion of dwell time and first fixations for faces in age groups 11-12, 13-14, and 15-16. For touched objects, females spent a larger proportion of dwell time in the age groups 5-6, 11-12, and 15-16, although no significant gender differences in first fixation proportions were found for any tested age group. Regarding bodies, females spent a larger proportion of dwell time in the age group 31-32, but no differences in first fixation proportions were found for any age group.

When analyzing gender differences pooled across all age groups, males exhibited larger dwell time and first fixation proportions towards faces. Conversely, females spent a larger proportion of their dwell time and first fixation proportions towards touched objects. Lastly, females spent a larger proportion of their dwell time on bodies, though no significant differences in first fixations were observed. Despite the statistical significance of these differences, effect sizes were very small (all $d < 0.24$). This is an important consideration as it indicates that while the differences are statistically significant in our very large sample, they may not be practically meaningful

These results are now included in the Supplementary Materials (pages 8-9, 1. 125-150):

Gender differences in semantic salience across age

Next, we examined gender differences in semantic salience. First, we calculated differences between males and females for each age group and semantic category included in our study. Figure S4 shows the age trajectories for semantic salience, indicated by dwell time proportion (A) and first fixation proportion (B) for males and females, respectively. Panel B displays the overall gender difference (pooled across age groups) across all included semantic metrics. Overall, our findings show similar developmental trajectories for male and female participants. We found no significant differences for text salience (all $p > .05$). For faces, male participants spent a

somewhat larger proportion of dwell time and first fixations in the age groups 11-12 (first fixation: $t(796) = 4.14, p < .01, d = 0.3$; dwell time: $t(796) = 4.85, p < .001, d = 0.35$) 13-14 (first fixation: $t(525) = 3.71, p < .01, d = 0.31$; dwell time: $t(525) = 6.69, p < .001, d = 0.59$), and 15-16 (first fixation: $t(309) = 4.03, p < .01, d = 0.47$; dwell time $t(309) = 4.27, p < .001, d = 0.49$). Regarding touched objects, males spent a smaller proportion of dwell time than females in the age groups 5-6 ($t(136) = -3.38, p < .05, d = -0.58$), 11-12 ($t(796) = -4.01, p < .01, d = -0.29$), and 15-16 ($t(309) = -3.2, p < .05, d = -3.7$). Males also spent a smaller proportion of dwell time on bodies in the age group 31-32 ($t(112) = -3.8, p < .01, d = -0.72$). All other age-wise comparisons were not significant (all $p > 0.05$). Finally, we examined gender differences in dwell time and first fixation proportions, pooled across all age groups. We observed no significant gender differences for text (first fixation: $t(6520) = 0.66, p = 0.51, d = 0.02$; dwell time $t(6520) = -0.98, p = 0.33, d = -0.02$). Males showed slightly higher face salience than females (first fixation: $t(6520) = 7.74, p < .001, d = 0.19$; dwell time: $t(6520) = 9.58, p < .001, d = 0.24$) and slightly lower salience for touched objects (first fixation: $t(6520) = -3.14, p < .01, d = -0.08$; dwell time $t(6520) = -5.17, p < .001, d = -0.13$). Lastly, we found that overall, males spent a smaller proportion of their dwell time ($t(6520) = -4.27, p < .001, d = -0.11$) on bodies, but did not differ from females in terms of first fixations towards bodies ($t(6520) = -1.93, p = 0.11, d = -0.05$), on bodies,.

Moreover, we refer to them in the results (page 8, l. 150-157):

Protracted development of gender differences

Previous social cognition research suggests females show a greater visual preference for social stimuli like faces compared to male participants (Lutchmaya & Baron-Cohen, 2002; Proverbio, 2017; Proverbio et al., 2008). Here, we explored gender differences in scene viewing. Surprisingly, males exhibited more dwell time and first fixations on faces in the 11-16 age range, while females focused more on touched objects at ages 5-6, 11-12 and 15-16. Pooled across age bins, males showed slightly higher gaze proportions towards faces and females towards touched objects and bodies (see Supplementary Materials for more details).

and discussion sections (page 17. l. 303-311):

Interestingly, we observed gender differences which pointed in the opposite direction of some previous studies (Lutchmaya & Baron-Cohen, 2002; Proverbio, 2017; Proverbio et al., 2008), with males showing slightly stronger face salience and females slightly stronger salience for touched objects and bodies. However, these differences were most pronounced in younger children and disappeared after teenage. Pooled across age groups, they had small effect sizes (all $d < .25$), suggesting limited practical significance (see Supplementary Materials). Nevertheless, these findings from our very large sample caution against the over-generalisation of gender differences observed in small samples. Additionally, the variation with age shows that gender

differences are modulated by further observer characteristics. Future work may probe whether culture is one of these.

Figure S4. Gender differences in semantic salience. Line plots display the mean proportion of dwell time (A) and first fixations (B) towards text, faces, touched objects and bodies (y-axis) across age bins from ages 5-6 to 71-72 (x-axis) for females (solid lines) and males (dotted lines). Please note that each age bin spans two years. Due to space constraints, only every second tick is labelled, and these are marked in bold for easier identification. Semantic categories are indicated by colours, as shown in the inset. Gray patches mark age bins with lower sensitivity to developmental trends ($n < 40$ per bin for females; $n < 50$ per bin for males). Panel C shows the mean proportion of dwell time and first fixations for females and males (desaturated colors) pooled across all age groups. The error bars indicates ± 1 standard error of the mean (SEM). Please note across all panels, the points above the lines indicate the level of significance for a given gender difference via shades of grey.

- I highly appreciate the contribution of the manuscript regarding the approximation of the metric tendencies using polynomial developmental trajectories. The approximative function itself may serve as a mathematical model for the specific metric and a visual stimulus set, being a function of the participant's age. In order to support this contribution of the manuscript, I would suggest evaluating the approximative functions on the validation set of participants to prove the presented results and support the conclusions along with the quantified error declaration.

We now performed cross-validations for 12 key metrics included in the present study by splitting our sample into two random halves: one for model fitting and the other for evaluation. Specifically, for each metric, we computed Pearson's correlations between the fitted age trajectories for one half of the participants and the observed bin-wise means of the other half. Additionally, we estimated noise ceilings for the fits by correlating the observed bin-wise means for both halves with each other.

The fitted age trajectories proved highly robust, with all cross-validated correlations between fitted curves and empirical means in the range of $r(25) = .73-.98$ (except first fixations towards bodies, for which we found no significant evidence for age-related changes in the first place, $r(25) = .54$). These correlations were all at the level of estimated noise ceilings or exceeded them, underscoring the robustness and validity of our curve-fitting approach. The new Figure S7 shows the correlation coefficients for each tested metric as well as the corresponding estimated noise ceilings and corresponding information is has been added to the Supplementary Materials (Page 13, lines 183-196):

Cross-validation of curve fitting

We cross-validated curve-fits by computing Pearson's correlations between developmental trajectories fitted to the data of odd numbered participants and the observed age-wise means for even numbered participants. Figure S7 shows the correlation coefficients for each tested metric as well as the corresponding estimated noise ceilings. We observed large cross-validated correlations between fitted developmental trajectories and hold-out data for dwell time proportions (text: $r(25) = 0.98$, $p < .001$; faces: $r(25) = 0.73$, $p < .001$; touched: $r(25) = 0.81$, $p < .001$; bodies: $r(25) = 0.73$, $p < .001$), first fixation proportions (text: $r(25) = 0.95$, $p < .001$; faces: $r(25) = 0.82$, $p < .001$; touched: $r(25) = 0.90$, $p < .001$; bodies: $r(25) = 0.54$, $p < .001$), as well as the center bias ($r(25) = 0.96$, $p < .001$), visual exploration ($r(25) = 0.92$, $p < .001$), bin-wise group-level entropy ($r(25) = 0.94$, $p < .001$) and the horizontal bias ($r(25) = 0.85$, $p < .001$). All cross-validated fit correlations were close to or exceeded their estimated noise ceilings (see Figure S7). Together, this indicates the robustness of the applied curve fitting.

Figure S7. Cross-validated correlations between fitted trajectories and hold-out data. Bar plots show cross-validated Pearson correlations for each key metric included in the study, as indicated by the tick labels. Horizontal lines with error shades show corresponding noise-ceiling estimates.

And referred to in the methods section (page. 25, l. 475-477):

We additionally confirmed the robustness of fits using cross-validation across independent subsamples of the data (see Supplementary Materials). The fitted parameters may inform future modelling studies (Kümmerer et al., 2016, 2017) and aid age-tailored approaches (Strauch et al., 2023).

- The authors state that the presentation order of the visual stimuli was constant for all the participants. This could introduce a great bias in the measured gaze data and, thus, the presented and discussed results. I would like to question the authors whether the results are invariant on the limited set of visual images and order of image dataset presented to the participants. Could the authors provide an additional validation eye-tracking research study on a smaller participant sample using natural scenes which were unseen in the study currently presented in the manuscript and evaluate the invariancy of the results on the used imagery dataset and the order of its presentation?

We appreciate this concern and now address it with additional analyses and data. Please note we kept all conditions constant, to maximise the comparability of data between different age groups and individuals.

We now tested the generalisability of all metrics used in the current study from our limited stimulus battery (OSIE 40) to a larger set of 660 scenes and to a different presentation order of the same 40 scenes. For this we reanalysed data from n = 103 adult observers who participated in a free viewing task in two separate sessions, about one week apart (see Linka & de Haas (2020) for more information). During the first session, participants viewed 700 images (Xu et al., 2014), with the initial 40 images being the ones used in our current study. On the second day of testing, the same participants viewed 200 images, starting with the 40 images we used, but this time displayed in a shuffled order.

Figure S8. Generalizability and shuffled retest reliability of individual differences to out of sample images. Bars and error bars show Pearson correlation coefficients with 95% confidence intervals for a given metric, as indicated by x-axis tick marks. (A) generalizability of estimates across the OSIE 40 and 660 out-of-sample images for objects across the semantic attributes text, faces, touched objects, and bodies, the center bias (mean eccentricity from image centroid), visual exploration (the number of distinct objects fixated), mean intra-individual entropy, and horizontal bias (measured by the proportion of horizontal saccades). (B) Corresponding Re-test reliabilities for the OSIE 40 images across different testing days and shuffled orders.. Note that all correlations here pertain to the robustness of individual estimates, and that of central tendencies will be substantially higher (see main text).

To examine the generalizability of gaze biases for our current stimulus set to those for other scenes, we calculated the correlation between individually estimated gaze parameters for our stimulus set versus those observed for the same participants viewing 660 other scenes (see above). The new figure S8B shows that even individual estimates based on the current stimulus set generalize to out-of-sample scenes ($r = .46-.75$, all $p < .001$). Again, these estimates pertain to the generalisability of *individual* estimates to out-of-sample scenes, implying a substantially higher robustness of *central tendencies* (like those of interest in our current study, see above).

To test a potential effect of image order, we tested the re-test reliability (Pearson's r) of individual gaze biases for the 40 stimuli used in our current

study across stimulus orders. Figure S8A shows the corresponding retest consistencies, which were all highly significant ($r = .33-.72$, all $p < .001$). Note that the lowest robustness estimates pertain to Bodies, for which we did not find developmental trends in our data. Again, please also note that these values indicate the robustness of *individual* estimates across stimulus orders. In the current study, we estimated the *central tendency* of age bins, which are considerably more robust, given the standard error of estimated means (s.e.m.) corresponds to a fraction of the variability at the individual level (determined by \sqrt{n}) and our large n for each age bin.

Crucially, we also tested the robustness of the reported age trajectories across independent stimuli in the current data ($n = 6,730$). We re-computed age trajectories for each metric, separately for odd and even scenes. Figure S9 displays the age trajectories for dwell time (A) and first fixation proportions (B) across all included semantic categories for both odd and even numbered images, respectively. Figure S10 presents the age trajectories for all visuo-spatial metrics for odd and even splits of the images, respectively.

Crucially, the observed developmental trajectories are highly consistent across subsets of scenes for all spatial biases and measures of exploration ($r = .86-.99$, all $p < .001$), as well as for dwell time and first fixations spent on text and faces ($r = .87-.97$, all $p < .001$). The cross-scene consistency of the trajectory for first fixations on touched objects was $r = .70$ ($p < .001$) and lower for dwell time ($r = .46$, $p < .05$). Importantly however, the developmental trajectory of dwell time on touched objects was highly robust in the age range for which we found developmental changes along this dimension in the first place (5-30 years; $r = .80$, $p < .001$). As expected, we found lower or no consistency for the development of dwell time ($r = -0.16$, $p = .42$). and first fixation tendencies ($r = .55$, $p < .01$) towards bodies, for which we found no systematic age-dependent variance in the first place.

Figure S9. Age trajectories for semantic salience for odd and even scenes Line plots display the mean proportion of dwell time (A) and first fixations (B) directed towards four semantic features across age bins for odd (left hand side) and even numbered scenes (right-hand side). Semantic categories are indicated by colours, as shown in the inset. Error shades indicate ± 1 standard error of the mean (SEM). Gray patches mark age bins with lower sensitivity to developmental trends ($n < 90$ per bin).

Figure S10. Age trajectories for visuo spatial metrics for odd and even. Line plots depict the average proportion of horizontal fixations, mean distance to the image centroid (in pixels), the average number of objects fixated, and intra-individual entropy for age bins ranging from ages 5-6 to 71-72 (x-axis) for even (left-hand side) and odd numbered scenes (right-hand side). Error shades indicate ± 1 standard error of the mean (SEM). Gray patches mark age bins with lower sensitivity to developmental trends ($n < 90$ per bin).

Together, these results provide evidence for the robustness of our findings to the specific selection and order of stimuli. We have included this information in the Supplementary Materials and refer to them in the method section (page 24, l. 454-456):

Additional control analyses demonstrated that all other key variables examined in this study can be reliably estimated using the OSIE40, regardless of image order (see Supplementary Materials).

as well as in the discussion section (page 20, l. 371-380):

Furthermore, we estimated semantic salience biases and visuo-spatial biases using the OSIE 40 free-viewing task, which includes 40 natural scenes presented in a serial order. This method may lack generalizability and could introduce order effects. To mitigate these risks, we validated the current stimulus battery (OSIE 40) in a previous, well-powered, and preregistered study (Linka & de Haas, 2020). This prior experiment explicitly tested the generalizability of individual gaze parameters for the OSIE 40 stimulus battery against a much larger set of scenes and confirmed the robustness of semantic gaze biases against stimulus order effects and their generalizability to out-of-sample scenes. We reanalysed this data regarding all metrics used in the current study and confirmed their robustness against effects of stimulus order and scene selection (see Supplementary Materials, Figure S8). We

further found that the observed developmental trajectories are consistent across independent subsets of scenes (see Supplementary Materials, Figures S9 and S10).

- The presented research study was designed with certain limitations, more specifically the number of natural scenes viewed by the participants, the specific order in which they were presented to the participants, and the various demographical and personal backgrounds of the participants resulting i.e. in biased top-down attention tendencies of the participants within and inter-group. The authors are encouraged to elaborate on the limitations of the design and measurements of their research study, as well as their impact on the results and conclusion statements presented in the manuscript. Moreover, I would encourage the authors to provide proof that their results and conclusions are invariant on the limitations of the design and the procedure of their research study and evaluation methodology.

We fully agree that addressing the limitations of the present study is important and have now added a limitations section to the discussion (Pages 19-20, l. 364-393). This section includes references to our control analyses (as discussed above) and provides context for interpreting our results in light of sample characteristics.

Limitations

The uneven age distribution of our sample resulted in limited sensitivity for developmental changes beyond age 56. At the same time, some variables (such as exploration tendency) appeared to develop across the entire life span. Also, we only included children from age 5 and previous research indicates that earlier periods represent a time of remarkable plasticity and significant cognitive and neural changes (Stiles, 2000). Future research could aim to track developmental changes more comprehensively, from infancy to old age. Furthermore, we estimated semantic salience biases and visuo-spatial biases using the OSIE 40 free-viewing task, which includes 40 natural scenes presented in a serial order. This method may lack generalizability and could introduce order effects. To mitigate these risks, we validated the current stimulus battery (OSIE 40) in a previous, well-powered, and preregistered study (Linka & de Haas, 2020). This prior experiment explicitly tested the generalizability of individual gaze parameters for the OSIE 40 stimulus battery against a much larger set of scenes and confirmed the robustness of semantic gaze biases against stimulus order effects and their generalizability to out-of-sample scenes. We reanalysed this data regarding all metrics used in the current study and confirmed their robustness against effects of stimulus order and scene selection (see Supplementary Materials, Figure S8). We further found that the observed developmental trajectories are consistent across independent subsets of scenes (see supplementary materials, Figures S9 and S10). Nevertheless, our results may not readily generalize to scenes or participants from an entirely different cultural background. Museum visitors from age 5-72 are a more diverse group of participants than typical samples of university students, but still far from representative of humanity. Both, our stimuli and participants likely are marked by western, educated, industrialized, rich and democratic (WEIRD, Henrich et al., 2010a) biases. Future studies could cross stimuli and participants from different cultural backgrounds, which would be a strong test case for the hypothesis that

developing gaze biases are linked to experience-dependent scene priors (see above).

Finally, our study employed a cross-sectional approach to examine developmental effects, which risks conflating age-related differences with generational effects and other demographic changes not accounted for in our data collection. Future longitudinal studies could trace individual developmental trajectories over time.

Minor remarks:

- In the abstract of the manuscript, lines 28-29, the authors state that the development of scene viewing is surprisingly protracted. However, the development of visual attention patterns during childhood and late adolescence is of state-of-the-art knowledge.

I would assume that the convergence of the visual attention patterns in the means of visual attractiveness of the socially affective stimuli towards the canonical state typical for adults is subject to protracted development discussed in the manuscript. Could the authors be more explicit about the facts in the manuscript?

Ref: Li, L., Gratton, C., Fabiani, M., & Knight, R. T. (2013). Age-related frontoparietal changes during the control of bottom-up and top-down attention: an ERP study. *Neurobiology of aging*, 34(2), 477-488.

Amso, D., Haas, S., & Markant, J. (2014). An eye tracking investigation of developmental change in bottom-up attention orienting to faces in cluttered natural scenes. *PloS one*, 9(1), e85701.

We respectfully disagree. Previous studies on the development of scene viewing either were group comparisons between children and adults or tried to infer developmental trajectories from a small number of groups with limited sample sizes. Crucially, these studies typically suggest that isolated visuospatial biases and features of oculomotor behavior (Helo et al., 2014; Krishna et al., 2018) or semantic biases such as face salience (Amso et al., 2014) reach an adult-like level around 8 years of age or earlier. Our study yields the first dataset allowing a comprehensive, well-powered and continuous tracing of scene viewing behavior from 5 years of age into adulthood. Compared to previous data and conclusions the resulting data indeed reveal surprisingly protracted development of semantic salience and basic spatial and oculomotor biases.

At the same time, we agree that our finding of protracted development of inter-observer coherence is particularly interesting (and reflecting the power afforded by massive sample sizes). We speculate it reflects the experience-dependent development of scene priors.

To communicate this more effectively, we have revised parts of the introduction and discussion. These revisions more explicitly relate and contrast our contribution to previous work and elaborate on our speculation regarding the development of scene priors.

Introduction (pages 3-4, l. 63-82):

*A central question in visual perception is whether and how oculomotor biases and attentional preferences develop. **Infants (Franchak et al., 2016) and children (Rider et al., 2018) watching movie clips exhibit greater***

interindividual variability than adults, suggesting that common attentional biases are learned. Although gaze preferences for faces and social information are present in infants (Andravizou et al., 2008; Gluckman & Johnson, 2013; Johnson et al., 1991, 2015; Kelly et al., 2019) and early childhood (Franchak et al., 2016; Frank et al., 2011), it remains poorly understood how they evolve from thereon. Amso et al., (2014) have reported the development of gaze towards faces to plateau around age 8 (Amso et al., 2014). Another study found lower text salience and higher salience of hands and objects being touched in preschoolers compared to adults (Linka et al., 2023). More basic patterns of oculomotor behaviour also seem to develop during childhood. Saccade amplitudes are higher and fixation durations are shorter at later stages of childhood (age 4-8) compared to age 2 (Helo et al., 2014) and the tendency to fixate on central image regions is weaker at age 8 than 4 (Krishna et al., 2018). The horizontal bias seems to be present in infants, albeit likely to a lesser extent than in adults (Van Renswoude et al., 2016). While most studies report the development of basic oculomotor behavior in children, other research finds no significant differences in visual exploration and fixation frequency between children and young adults (Acik et al., 2010). Together, these studies suggest that basic visuospatial and semantic salience biases differ between young children and adults but may mature as early as age 8. Overall, however our understanding of gaze development is severely limited by a scarcity of data, with most studies comparing two or three age groups with limited sample sizes, which rarely cover adolescence and do not allow a continuous comparison of gaze across development. It thus remains unclear whether and how scene viewing changes beyond early childhood.

Page 4, l. 95-97:

Here, we aimed to test whether scene viewing matures early or develops into adolescence and beyond. Therefore, we traced the developmental trajectories of semantic salience and basic oculomotor biases continuously and across a wide age range

Page 5, l. 109-113:

We find strong evidence for a protracted development of attentional preferences and basic oculomotor biases extending into teenage and young adulthood. Furthermore, children viewed scenes in more idiosyncratic ways, which gradually gave way to more canonical patterns, which fully matured in the mid-20s.

Discussion (pages 14-15, l. 234-241):

Previous studies using small and focussed samples have reported differences in semantic salience (Amso et al., 2014; Franchak et al., 2016; Linka et al., 2023) and facets of oculomotor behaviour (Helo et al., 2014; Krishna et al., 2018) between adults and children younger than eight years of age. Here, we found surprisingly protracted development for the semantic salience of text, faces, and touched objects, as well as for visual exploration and basic

oculomotor biases, such as the central and horizontal bias, all of which continuously developed into adolescence and for some measures up until young adulthood. Inter-observer similarity steeply increased throughout teenage and only plateaued in the late-20s. This suggests that typical adult gaze behaviour takes two decades to unfold and requires prolonged visual experience.

Discussion (pages 18-19, l. 341-363):

Protracted development of scene priors?

A common thread across our findings is the potential role of visual experience and understanding. We speculate that the decreasing salience of touched objects may be driven by the developing ability to grasp the global meaning of (inter)actions from scene gist, without the need to foveate hands. The protracted increase of text salience likely is linked to education and an increasing share of text elements in the foveal diet. The horizontal bias may unfold along with spatial priors about typical object arrangements in everyday scenes. Most importantly, the unfolding canonical nature of scene exploration may point to the development of common templates for everyday scenes and their most relevant or informative regions. Individual visual diets may become less idiosyncratic from the onset of out-of-home care (Jayaraman & Smith, 2019). Future studies could use mobile eyetracking to test whether the intraindividual variance of visual diets increases with age, while the inter-individual variance decreases. Future studies may further compare the cultural specificity of canonical viewing behavior at the group level. If culturally specific priors shape scene viewing, the decreasing variance among adults with a similar cultural background should be contrasted by increasing inter-cultural differences when comparing groups with different backgrounds. Interestingly, we observed protracted development for semantic dwell-time salience, as well as for (distinct) semantic biases of the immediate saccade after image onset, and low-level oculomotor features like the horizontal bias. Therefore, both, bottom-up and top-down aspects of processing likely are subject to protracted development^{32,35,37} and may be shaped by a self-reinforcing interplay between unfolding scene priors and active vision³⁷. Future research may use different tasks and neuroimaging to disentangle the role of bottom-up and top-down processing in the development of scene viewing.

- The introduction of the manuscript (lines 53-61) discusses visual attention tendencies regarding low-level and high-level features of natural scenes mainly from the point of view of bottom-up attention mechanisms. This should be explicitly stated in the manuscript. Moreover, I would suggest not neglecting the top-down attention mechanisms and their relation with the observed gaze behaviour, mainly the internal guidance originating in the internal state of the observer, memory, etc.

Ref: Theeuwes, J. (2010). Top-down and bottom-up control of visual selection. *Acta psychologica*, 135(2), 77-99.

Baluch, F., & Itti, L. (2011). Mechanisms of top-down attention. *Trends in neurosciences*, 34(4), 210-224.

We thank the reviewer for pointing this out. We agree that it is necessary to embed our work within the broader context of research on visual attention. We now state in the introduction that there are two classic lines of work studying visual attention in human observers: those focusing on stimulus-driven, bottom-up influences and those focusing on top-down influences driven by observer states and task demands. We also highlight recent work suggesting that free-viewing behaviour towards complex scenes is driven by a ‘default’ task of extracting global meaning. We additionally discuss that developmental changes may reflect change in both, stimulus-driven bottom-up processing and internal top-down priors (see above), as well as their interplay (page 3, l. 55-59).

When predicting free viewing behaviour, higher-level features, like Faces, Text, Touched objects and Taste (Wang & Pomplun, 2012; Xu et al., 2014) carry more weight than their lower-level counterparts (Stoll et al., 2015; Xu et al., 2014). The extraction of global meaning likely is a main objective of scene viewing without additional task demands (Murlidaran & Eckstein, 2024). Besides bottom-up stimulus drives, viewing behaviour can be shaped by top-down task demands and internal states (Hayhoe et al., 2005; Hayhoe & Ballard, 2005; Theeuwes, 2010) and traits (Haas et al., 2018; Linka & de Haas, 2020 ;Constantino et al., 2017; Kennedy et al., 2017) of the observer

We further discuss the potential implications of our findings for top-down control of gaze behavior and how future research might build on this in the discussion section on page 19, l. 357-363:

Interestingly, we observed protracted development for semantic dwell-time salience, as well as for (distinct) semantic biases of the immediate saccade after image onset, and low-level oculomotor features like the horizontal bias. Therefore, both, bottom-up and top-down aspects of processing likely are subject to protracted development (Acik et al., 2010; Amso et al., 2014; Krishna et al., 2018) and may be shaped by a self-reinforcing interplay between unfolding scene priors and active vision. Future research may use different tasks and neuroimaging to disentangle the role of bottom-up and top-down processing in the development of scene viewing.

- The authors suggest evaluating various metrics from 3-second long periods of image observation sequences to depict the gaze development phenomena in the manuscript. However, I would appreciate the authors explicitly stating the interconnection of the metrics proposal with the broader context of visual perception and visual attention strategies depicted in the manuscript in lines 62-87 and then in the Discussion section to point out the tested hypotheses early in the proposal.

We appreciate the author's feedback. We have revised parts of the introduction to better motivate the metrics used and to explicitly state our research question early on. Additionally, we have addressed the research question early in the discussion section.

Introduction (pages 3-4, l.77-82):

Together, these studies suggest that basic visuospatial and semantic salience biases differ between young children and adults, but may mature as early as

age 8. Overall, however our understanding of gaze development is severely limited by a scarcity of data, with most studies comparing two or three age groups with limited sample sizes, which rarely cover adolescence and do not allow a continuous comparison of gaze across development. It thus remains unclear whether and how scene viewing changes beyond early childhood.

And page 4, l. 92-97:

Together, these studies suggest that basic aspects of visual processing continue to evolve into adolescence and raise the possibility that this extends to oculomotor biases and attentional preferences.

Here, we aimed to test whether scene viewing matures early or develops into adolescence and beyond. Therefore, we traced the developmental trajectories of semantic salience and basic oculomotor biases continuously and across a wide age range.

Discussion (pages 14-15, l. 233-241)

To probe the developmental trajectory of free-viewing behaviour towards natural scenes, we analysed the gaze of thousands of individuals from 5 to 72 years of age. Previous studies using small and focussed samples have reported differences in semantic salience (Amso et al., 2014; Franchak et al., 2016; Linka et al., 2023) and facets of oculomotor behaviour (Helo et al., 2014; Krishna et al., 2018) between adults and children younger than eight years of age. Here, we found surprisingly protracted development for the semantic salience of text, faces, and touched objects, as well as for visual exploration and basic oculomotor biases, such as the central and horizontal bias, all of which continuously developed into adolescence and for some measures up until young adulthood. Inter-observer similarity steeply increased throughout teenage and only plateaued in the late-20s

- The polynomial developmental trajectory displayed in Figure 1/B does not reflect the absent evidence of change for ages over 56 and seems to wrongly suggest that the feature proportion increases for the elderly population. I assume this is the case also for Figure 1/A, Figure 2/B, Figure 3/B,C and Figure 4/A,D. I would suggest adjusting the polynomial developmental trajectory data visualization for the particularly non-sensitive age ranges (as stated in line 97).

We fully agree. We have now excluded age bins over 56 for the curve fitting analyses. Further, we have added shaded regions to each line plot to highlight the age ranges where our statistical power is limited and refer to this in the figure caption (see below for an example figure).

Figure 4. Protracted development of inter-observer similarity. Panel A shows the average entropy of group-level fixation maps across age bins. Panel B shows fixation data from observers aged 5-22 overlaid as colored circles on one example image. Colors correspond to age groups as indicated by the colorbar. Note the lower dispersion of hotter colors, indicating less dispersion on the group level for adult participants. Panel C shows the average entropy of observer-specific fixation maps across age bins. Panel D shows the mean observer similarity for each age bin, as indicated by the average pairwise correlation of observer-specific dwell-time distributions across objects. **Note that for panels A, C and D, each age bin spans two years. Due to space constraints, only every second tick is labelled, and these are marked in bold for easier identification. Gray patches mark age bins with lower sensitivity to developmental changes, which were disregarded for curve fitting ($n < 90$ per bin).** All error shades indicate ± 1 standard error of the mean (SEM) and the dotted lines the best fitting polynomial. The lines above the x-axes indicate the level of evidence for a given bin to be part of broad linear change via shades of grey, as shown in the inset (sliding window analysis, cf. main text and methods).

- In line 353, the authors state that the information about previous participation, and wearing glasses was collected from the participants. Was the participant able to be involved in the study if previously participated? Was the participant able to be involved in the study if wearing glasses? If yes, could this affect the results?

We thank the reviewer for pointing this out. Visitors had the opportunity to participate in the study multiple times. For all analyses reported in the present study, we excluded data from participants who indicated prior participation. We now make this clearer in the methods section on page 21, l. 404-406:

10632 of these stemmed from individual participants, *who indicated no prior participation* and 10624 of these proceeded to the free viewing task (cf. Figure 6 B)

Since the manufacturer asserts that their eye-tracking system is robust against the use of glasses, we did not exclude participants who wore glasses during participation (unless they failed our stringent accuracy thresholds for inclusion, see above). Additional analyses excluding all participants who reported wearing glasses replicate all findings (see supplementary results and figure below).

Robustness of gaze trajectories to participants wearing glasses

To probe whether the reported age trajectories are affected by participants wearing glasses, we recalculated developmental trajectories for all key metrics omitting participants who reported wearing glasses ($n = 1922$). Figure S12 shows age trajectories for dwell time (A) and first fixation proportions (B) across all included semantic object dimensions, and for visuospatial gaze biases (C). All results of the main analysis closely replicated.

Figure S12. Developmental trajectories excluding participants wearing glasses. Mean proportion of dwell time (A) and first fixations (B) directed towards objects of four semantic features across age bins ranging from ages 5-6 to 71-72 (x-axis) when excluding data from participants who indicated wearing glasses. Panel C shows the average proportion of horizontal fixations, mean distance to image centroid (in pixels), average number of objects fixated, and intra-individual entropy for age bins (x-axis) when excluding participants who indicated wearing glasses. Error shades indicate ± 1 standard error of the mean (SEM), dotted lines best polynomial fits. The lines above the x-axes indicate the level of evidence for a given bin to be part of broad linear change via shades of grey, as shown in the inset (sliding window analysis, cf. main text and methods). Please note that each age bin spans two years. Due to space constraints, only every second tick is labelled, and these are marked in bold for easier identification. Gray patches mark age bins with lower sensitivity to developmental trends ($n < 30$ per bin).

- The authors validated the accuracy of the used eye-tracker and compared it to the predecessor eye-tracker version (lines 342-344). Could the authors provide information about the methodology of the validation practice trials executed?

Please note that we used the Tobii 4c referred to here (and did not compare it to any predecessor or successor model). We piloted the calibration and validation procedure in the lab before implementing it in the museum. In line with previous reports on the quality of the 4c, we routinely achieved accuracies well below 0.6 dva with the 9-point calibration/validation procedure with the Eye Tracker Manager interface developed by Tobii. Additionally, we generated trace plots to visualize the fixations overlaid on the images for plausibility (see supplementary materials, Figure S11).

- Does visualization in Figure 1 in the form of fixation map image overlay correctly display the eye-tracking measurement error quantified from the measured data? If yes, how the sigma of the Gaussian kernels for the plotted fixations was calculated for visualization purposes and also for fixation map smoothing to compute Shannon entropy (line 186)?

The two overlay plots (middle and right-hand side) in Panel C show fixations represented as alpha-blended disks with an individual extent corresponding to about 0.5 dva (which is roughly equivalent to average data accuracy). To smooth the fixation maps for computing Shannon entropy, we used a sigma (standard deviation) of 1 degree of visual angle (dva), matching the measurement error threshold of <1 dva used as an inclusion criterion for further analyses. We now mention this on page 26, l. 404-406:

A given fixation map was created by applying a Gaussian filter with a standard deviation of 1 dva to each fixation.

- In lines 352-353, the authors state the participants provided informed consent to participate in the study. How was this ensured regarding participants under 18 years for whom the consent should originate from their parent or legal representative?

The automated procedure indeed requested informed consent, and in the case of minors, it asked for consent from their legal representative, in line with IRB requirements (please note minors typically do not visit the museum unattended). We have now added this information to the methods section on Page 22, lines 413-414.

For minors, the procedure requested consent from a legal representative.

- Could the authors give the reasoning for choosing namely the NH2010 fixation/saccadic detection algorithm among the others available?

Ref: Hooge I., et al. (2022). Fixation classification: how to merge and select fixation candidates. Behavior Research Methods. 54. 10.3758/s13428-021-01723-1.

This algorithm is easily accessible and implementable in MATLAB and is free to use. We also tested other fixation extraction algorithms, including custom in-house solutions, and found that the NH2010 algorithm performed better when manually inspecting overlays of gaze traces. Nevertheless, it can have a conservative tendency to classify nearby fixations separated by very small saccades as single fixations, which contributes to the comparably low fixation frequencies overall. We now briefly point this out in the results (page. 12, l. 186-189) and highlight example trace plots and classification in Figure S11

Overall, estimated fixation frequencies were conservative, given the event classification procedure could label fixations separated by very small saccades as single fixations (see supplementary Figure S11 for examples). Reassuringly, the developmental trend for increasing exploration replicated in the number of fixated objects across all images,

Reviewer #3:

Remarks to the Author:

This study examines the development of gaze behavior on natural scenes. The study investigates several characteristics of gaze and compares them across age groups from early childhood to older age. It does so with an unusually large cohort of participants (6736). The study seems to be exploratory in its approach, as it examines a few different variables, without presenting a hypotheses or prediction to tie them together. I find a lot of value in this exploratory approach and in the large dataset (which I am happy to see, will be made publicly available). However, taking all the findings together, I am left unsure of what is the take-home message.

The main claim that the study is making is that gaze development is protracted, with some characteristics developing over the first two decades of life. I have a few concerns regarding this conclusion:

a. I think that for some of the measurements this conclusion derives from the polynomial curve fitting that was used. Looking at some of the figures, it seems as though the measurements reach a plateau earlier than the plateaus of the fitted function (which may suggest that the function is not the right fit). For example, in Figure 1A (touched and faces), in Figure 2B and in Figure 3 A+B, the data seems to plateau at early teens. While the fitted

function suggests later. In figure 2B it can be clearly seen how the fitted function “forces” a developmental trend from late teens to early 30s. The collected data is higher than the fitted function for the early teens and then becomes lower for the 20s and until early 30s. This may create the illusion of a slow development, when in fact the plateau has been reached earlier than it seems (from around age 15, there doesn’t seem to be any increase).

Thank you for pointing this out. Our approach to quantifying changes for a given metric across age was twofold. First, we computed the fit of the global trend to minimize subjectivity. Following the suggestion of reviewer 2, we have now excluded age bins > 56 for this, which resulted in improved fits, particularly for the examples flagged as concerning here. Please see the updated Figure 2B below as an example (and also above for the cross-validation analyses showing the robustness and validity of our approach).

Our second approach was to compute the evidence for change using moving windows (size 5 bins). We projected the mean evidence for change across all

windows containing a given age bin and projected this value onto the bin. This process effectively smoothes the data and may render the time-window of apparent development more protracted than it actually is, as pointed out by the reviewer.

We have now recalculated all analyses using a much more conservative approach. We evaluate the evidence for change across any two consecutive age bins using a single window which has these two age bins as leftmost entries (total size still 5 bins). This is the least protracted interpretation possible - developmental change within a window corresponds *at least* to change across its earliest two entries. This highly conservative approach is more in line with visual inspection of the data and pushes some developmental inflection points back to early adulthood or teenage, which is reflected throughout the updated manuscript.

b. The introduction and the discussion do not tell a full story on what was previously known and what is currently revealed. I understand that the conclusion is that development is slow and lasts for two decades, but it remains unclear if this contradicts the current consensus. The findings are described one after the other, lacking a coherent theory or story to explain them and lacking the background to reveal whether (and if yes, why) they are groundbreaking.

We thank the reviewer for the feedback. We now outline the state of the art and our contribution more clearly (pages 3-4, l. 63-82):

A central question in visual perception is whether and how oculomotor biases and attentional preferences develop. Infants (Franchak et al., 2016) and children (Rider et al., 2018) watching movie clips exhibit greater interindividual variability than adults, suggesting that common attentional biases are learned. Although gaze preferences for faces and social information are present in infants (Andravizou et al., 2008; Gluckman & Johnson, 2013; Johnson et al., 1991, 2015; Kelly et al., 2019) and early childhood (Franchak et al., 2016; Frank et al., 2011), it remains poorly understood how they evolve from thereon. Amso et al., (2014) have reported the development of gaze towards faces to plateau around age 8 (Amso et al., 2014). Another study found lower text salience and higher salience of hands and objects being touched in preschoolers compared to adults (Linka et al., 2023). More basic patterns of oculomotor behaviour also seem to develop during childhood. Saccade amplitudes are higher and fixation durations are shorter at later stages of childhood (age 4-8) compared to age 2 (Helo et al., 2014) and the tendency to fixate on central image regions is weaker at age 8 than 4 (Krishna et al., 2018). The horizontal bias seems to be present in infants, albeit likely to a lesser extent than in adults (Van Renswoude et al., 2016). While most studies report the development of basic oculomotor behavior in children, other research finds no significant differences in visual exploration and fixation frequency between children and young adults (Acik et al., 2010). Together, these studies suggest that basic visuospatial and semantic salience biases differ between young children and adults but may mature as early as age 8. Overall, however our understanding of gaze development is severely limited by a scarcity of data, with most studies comparing two or three age groups with limited sample sizes, which rarely

cover adolescence and do not allow a continuous comparison of gaze across development. It thus remains unclear whether and how scene viewing changes beyond early childhood.

Introduction (Page 5. l. 95-97):

Here, we aimed to test whether scene viewing matures early or develops into adolescence and beyond. Therefore, we traced the developmental trajectories of semantic salience and basic oculomotor biases continuously and across a wide age range

And page 5, l. 109-113:

We find strong evidence for a protracted development of attentional preferences and basic oculomotor biases extending into teenage and young adulthood. Furthermore, children viewed scenes in more idiosyncratic ways, which gradually gave way to more canonical patterns, which fully matured in the mid-20s.

Discussion (Pages 14-15, l. 234-242):

Previous studies using small and focussed samples have reported differences in semantic salience (Amso et al., 2014; Franchak et al., 2016; Linka et al., 2023) and facets of oculomotor behaviour (Helo et al., 2014; Krishna et al., 2018) between adults and children younger than eight years of age. Here, we found surprisingly protracted development for the semantic salience of text, faces, and touched objects, as well as for visual exploration and basic oculomotor biases, such as the central and horizontal bias, all of which continuously developed into adolescence and for some measures up until young adulthood. Inter-observer similarity steeply increased throughout teenage and only plateaued in the late-20s. This suggests that typical adult gaze behaviour takes two decades to unfold and requires prolonged visual experience.

Additionally, we have expanded and restructured the discussion section to emphasise a speculative, albeit coherent interpretation of our various findings, namely the role of prior visual knowledge (page 15. l. 244-246).

In the following, we will discuss each of our findings in detail and the hypothesis that protracted gaze development reflects a gradual accumulation of priors about the visual world.

And pages 18-19. l. 341-363

Protracted development of scene priors?

A common thread across our findings is the potential role of visual experience and understanding. We speculate that the decreasing salience of touched objects may be driven by the developing ability to grasp the global meaning of (inter)actions from scene gist, without the need to foveate hands. The

protracted increase of text salience likely is linked to education and an increasing share of text elements in the foveal diet. The horizontal bias may unfold along with spatial priors about typical object arrangements in everyday scenes. Most importantly, the unfolding canonical nature of scene exploration may point to the development of common templates for everyday scenes and their most relevant or informative regions. Individual visual diets may become less idiosyncratic from the onset of out-of-home care (Jayaraman & Smith, 2019). Future studies could use mobile eyetracking to test whether the intraindividual variance of visual diets increases with age, while the inter-individual variance decreases. Future studies may further compare the cultural specificity of canonical viewing behavior at the group level. If culturally specific priors shape scene viewing, the decreasing variance among adults with a similar cultural background should be contrasted by increasing inter-cultural differences when comparing groups with different backgrounds. Interestingly, we observed protracted development for semantic dwell-time salience, as well as for (distinct) semantic biases of the immediate saccade after image onset, and low-level oculomotor features like the horizontal bias. Therefore, both, bottom-up and top-down aspects of processing likely are subject to protracted development (Acik et al., 2010; Amso et al., 2014; Krishna et al., 2018) and may be shaped by a self-reinforcing interplay between unfolding scene priors and active vision. Future research may use different tasks and neuroimaging to disentangle the role of bottom-up and top-down processing in the development of scene viewing.

In addition I have a few minor comments:

a. I might have missed it, but I can't find a summary of participants numbers in the different age groups, or their distribution across the age group. This is very important, and I think would be best if it comes as a figure. It's also important to know what the modulation of variability across the groups are.

We agree and therefore did provide a figure with a histogram in the methods section (Figure 5; see also below), which may have slipped the reviewers attention. Regarding variability, all developmental traces show ± 1 S.E.M. as error shades in a bin-wise fashion.

Figure 5. Age distribution of analysed free viewing data. Histogram displaying the distribution of ages of included observers using a bin width of two.

b. Page 16, line 207 missing “and”

Thank you, fixed

c. The number of participants is not reported consistently. It appears in the abstract as >6500. Then in the introduction it says “over 10,000” and in fact it was 6736

We gathered data from more than 10,000 participants, but included 6,736 high quality datasets. We now make this clearer from the start. In the abstract (page 2, l. 28-29):

Here, we present data from $n = 6730$ subjects from 5-72 years of age, freely viewing 40 natural scenes

And introduction (page 4, l. 97-100):

*In collaboration with a local science museum, we gathered gaze data from over 10,000 participants ranging from 5 to 72 years of age. **Out of these, we selected 6,730 participants for further analysis based on the high quality of their data.***

d. The sentence “fundamental aspects of adult gaze take decades of continuous development” from the abstract is focusing. It sounds like development “takes decades” when in fact it is around two decades at the most. So this may be misleading.

We agree and have adapted this part (page 2, l. 35-37)

*These findings show that fundamental aspects of adult gaze **take up to two decades** of continuous development and push individuals towards more canonical viewing patterns.*

References

- Acik, A., Sarwary, A., Schultze-Kraft, R., Onat, S., & König, P. (2010). Developmental Changes in Natural Viewing Behavior: Bottom-Up and Top-Down Differences between Children, Young Adults and Older Adults. *Frontiers in Psychology*, 1, 207. <https://doi.org/10.3389/fpsyg.2010.00207>
- Amso, D., Haas, S., & Markant, J. (2014). An Eye Tracking Investigation of Developmental Change in Bottom-up Attention Orienting to Faces in Cluttered Natural Scenes. *PLOS ONE*, 9(1), e85701-. <https://doi.org/10.1371/journal.pone.0085701>
- Andravizou, A., Gliga, T., Elsabbagh, M., & Johnson, M. (2008). Faces Attract Infants' Attention in Complex Displays. *Annals of General Psychiatry*, 14. <https://doi.org/10.1186/1744-859X-7-S1-S276>
- Constantino, J. N., Kennon-McGill, S., Weichselbaum, C., Marrus, N., Haider, A., Glowinski, A. L., Gillespie, S., Klaiman, C., Klin, A., & Jones, W. (2017). Infant viewing of social scenes is under genetic control and is atypical in autism. *Nature*, 547(7663), 340–344. <https://doi.org/10.1038/nature22999>
- Franchak, J. M., Heeger, D. J., Hasson, U., & Adolph, K. E. (2016). Free Viewing Gaze Behavior in Infants and Adults. *Infancy*, 21(3), 262–287. <https://doi.org/https://doi.org/10.1111/infa.12119>
- Frank, M., Vul, E., & Saxe, R. (2011). Measuring the Development of Social Attention Using Free-Viewing. *Infancy*, 17. <https://doi.org/10.1111/j.1532-7078.2011.00086.x>
- Gluckman, M., & Johnson, S. (2013). Attentional capture by social stimuli in young infants. *Frontiers in Psychology*, 4, 527. <https://doi.org/10.3389/fpsyg.2013.00527>
- Haas, B., Iakovidis, A., Schwarzkopf, D., & Gegenfurtner, K. (2018). Individual differences in visual salience vary along semantic dimensions. <https://doi.org/10.1101/444257>
- Hayhoe, M., & Ballard, D. (2005). Eye movements in natural behavior. *Trends in Cognitive Sciences*, 9(4), 188–194.
- Hayhoe, M., Mennie, N., Sullivan, B., & Gorgos, K. (2005). The role of internal models and prediction in catching balls. *Proceedings of the American Association for Artificial Intelligence*, 1–5.
- Helo, A., Pannasch, S., Sirri, L., & Rämä, P. (2014). The maturation of eye movement behavior: Scene viewing characteristics in children and adults. *Vision Research*, 103, 83–91. <https://doi.org/10.1016/j.visres.2014.08.006>
- Henrich, J., Heine, S. J., & Norenzayan, A. (2010a). Most people are not WEIRD. *Nature*, 466(7302), 29. <https://doi.org/10.1038/466029a>
- Henrich, J., Heine, S. J., & Norenzayan, A. (2010b). The weirdest people in the world? *Behavioral and Brain Sciences*, 33(2–3), 61–83. <https://doi.org/DOI:10.1017/S0140525X0999152X>

- Jayaraman, S., & Smith, L. B. (2019). Faces in early visual environments are persistent not just frequent. *Vision Research*, 157, 213–221.
<https://doi.org/https://doi.org/10.1016/j.visres.2018.05.005>
- Johnson, M. H., Dziurawiec, S., Ellis, H., & Morton, J. (1991). Newborns' preferential tracking of face-like stimuli and its subsequent decline. *Cognition*, 40(1), 1–19.
[https://doi.org/https://doi.org/10.1016/0010-0277\(91\)90045-6](https://doi.org/https://doi.org/10.1016/0010-0277(91)90045-6)
- Johnson, M. H., Senju, A., & Tomalski, P. (2015). The two-process theory of face processing: Modifications based on two decades of data from infants and adults. *Neuroscience & Biobehavioral Reviews*, 50, 169–179.
<https://doi.org/https://doi.org/10.1016/j.neubiorev.2014.10.009>
- Kelly, D., Duarte, S., Meary, D., Bindemann, M., & Pascalis, O. (2019). Infants rapidly detect human faces in complex naturalistic visual scenes. *Developmental Science*, 22.
<https://doi.org/10.1111/desc.12829>
- Kennedy, D. P., D'Onofrio, B. M., Quinn, P. D., Bölte, S., Lichtenstein, P., & Falck-Ytter, T. (2017). Genetic Influence on Eye Movements to Complex Scenes at Short Timescales. *Current Biology*, 27(22), 3554–3560.e3.
<https://doi.org/https://doi.org/10.1016/j.cub.2017.10.007>
- Krishna, O., Helo, A., Rämä, P., & Aizawa, K. (2018). Gaze distribution analysis and saliency prediction across age groups. *PLOS ONE*, 13(2), e0193149-
<https://doi.org/10.1371/journal.pone.0193149>
- Kümmerer, M., Wallis, T. S. A., & Bethge, M. (2016). DeepGaze II: Reading fixations from deep features trained on object recognition. *CoRR*, abs/1610.01563.
<http://arxiv.org/abs/1610.01563>
- Kümmerer, M., Wallis, T. S. A., Gatys, L. A., & Bethge, M. (2017). Understanding Low- and High-Level Contributions to Fixation Prediction. 2017 IEEE International Conference on Computer Vision (ICCV), 4799–4808. <https://doi.org/10.1109/ICCV.2017.513>
- Linka, M., & de Haas, B. (2020). OSIEshort: A small stimulus set can reliably estimate individual differences in semantic salience. *Journal of Vision*, 20(9), 13.
<https://doi.org/10.1167/jov.20.9.13>
- Linka, M., Sensoy, Ö., Karimpur, H., Schwarzer, G., & de Haas, B. (2023). Free viewing biases for complex scenes in preschoolers and adults. *Scientific Reports*, 13(1), 11803.
<https://doi.org/10.1038/s41598-023-38854-8>
- Lutchmaya, S., & Baron-Cohen, S. (2002). Human sex differences in social and non-social looking preferences, at 12 months of age. *Infant Behavior and Development*, 25(3), 319–325. [https://doi.org/https://doi.org/10.1016/S0163-6383\(02\)00095-4](https://doi.org/https://doi.org/10.1016/S0163-6383(02)00095-4)
- Murlidaran, S., & Eckstein, M. P. (2024). Eye Movements during Free Viewing Maximize Scene Understanding.
- Proverbio, A. M. (2017). Sex differences in social cognition: The case of face processing. *Journal of Neuroscience Research*, 95(1–2), 222–234.
<https://doi.org/https://doi.org/10.1002/jnr.23817>

- Proverbio, A. M., Zani, A., & Adorni, R. (2008). Neural markers of a greater female responsiveness to social stimuli. *BMC Neuroscience*, 9(1), 56. <https://doi.org/10.1186/1471-2202-9-56>
- Rider, A., Coutrot, A., Pellicano, E., Dakin, S., & Mareschal, I. (2018). Semantic content outweighs low-level saliency in determining children's and adults' fixation of movies. *Journal of Experimental Child Psychology*, 166, 293–309. <https://doi.org/10.1016/j.jecp.2017.09.002>
- Stiles, J. (2000). Neural plasticity and cognitive development. *Developmental Neuropsychology*, 18, 237–272.
- Stoll, J., Thrun, M., Nuthmann, A., & Einhäuser, W. (2015). Overt attention in natural scenes: Objects dominate features. *Vision Research*, 107, 36–48. <https://doi.org/https://doi.org/10.1016/j.visres.2014.11.006>
- Strauch, C., Hoogerbrugge, A. J., Baer, G., Hooge, I. T. C., Nijboer, T. C. W., Stuit, S. M., & Van der Stigchel, S. (2023). Saliency models perform best for women's and young adults' fixations. *Communications Psychology*, 1(1), 34. <https://doi.org/10.1038/s44271-023-00035-8>
- Theeuwes, J. (2010). Top–down and bottom–up control of visual selection. *Acta Psychologica*, 135(2), 77–99. <https://doi.org/https://doi.org/10.1016/j.actpsy.2010.02.006>
- Van Renswoude, D. R., Johnson, S. P., Raijmakers, M. E. J., & Visser, I. (2016). Do infants have the horizontal bias? *Infant Behavior and Development*, 44, 38–48. <https://doi.org/https://doi.org/10.1016/j.infbeh.2016.05.005>
- Wang, H.-C., & Pomplun, M. (2012). The attraction of visual attention to texts in real-world scenes. *Journal of Vision*, 12. <https://doi.org/10.1167/12.6.26>
- Xu, J., Jiang, M., Wang, S., Kankanhalli, M., & Zhao, Q. (2014). Predicting human gaze beyond pixels. *Journal of Vision*, 14. <https://doi.org/10.1167/14.1.28>

Reviewer #1 (Remarks to the Author):

Thank you for your response. While it has addressed some of my concerns, I remain uncertain about the approach to curve fitting.

I have carefully read the response and the manuscript, the curve fitting was applied to several metrics with age as the variable. However, the author does not specify whether the input values for age were integers or decimals. I assume they were integers (e.g., 7 years old rather than 7.5). If that assumption is correct, I question the necessity of polynomial fitting, which seems to serve only to enhance the visualization. For instance, if we want to estimate the dwell time for 8-year-olds, using the average value for all individuals aged 8 would likely be more reliable than relying on a value derived from the curve fitting. Moreover, the absence of decimal data reduces the model's confidence in estimating values for intermediate ages, such as 7.5 years.

We thank the reviewer for highlighting this oversight. We have now included this information in our methods (p. 24, l. 469-471):

Subsequently, they provided informed consent, indicated their age (as integers), gender, prior participation (y/n), and whether they wore glasses (y/n).

As previously outlined in the Methods (p. 26-27), observed averages pertain to age bins spanning two years, ensuring well-powered samples from 5-6 to 55-56 years of age. We now made this more salient in the introduction section as well (p. 5, l. 109-110):

We calculated these averages individually for age bins spanning two years each, ranging from 5-6 to 71-72 years.

And discuss age discretization in the limitations section on p. 20, l. 395-399:

To enhance the sample size per age group and the robustness of related point estimates, all fitted age trajectories and curve fits are based on age bins spanning 2 years of age. Future research may conduct experiments at an even larger scale, or focus on smaller age ranges to trace developmental trajectories at a higher resolution (e.g. age in months).

Regarding our plots, first, we agree that the observed averages are central and should be clearly visible. We previously plotted observed averages with linearly interpolated solid lines, which we now realise lacked clarity. Therefore, we updated all plots to show bin-wise averages as scatters with error bars rendering them clearly discernible. See Fig. 2 below as an example.

Figure 2. Protracted development of the horizontal bias. Panel A shows polar saccade plots across 36 direction bins for ages 5-6 to 13-14. Age is indicated by colour as shown in the inset. Panel B shows the average proportion of horizontal fixations across age bins \pm 1 SEM (scatter points and error bars, respectively). Please note that each age bin spans two years. Due to space constraints, only every second tick is labelled, and these are marked in bold for easier identification. The black line shows the best fitting polynomial. The horizontal lines at the top indicate the level of evidence for a given bin to be part of broad linear change via shades of grey, as shown in the inset (sliding window analysis, cf. main text and methods). The red patch marks age bins with lower sensitivity for developmental changes, which were disregarded for curve fitting ($n < 90$ per bin).

As for the fitted polynomials, we still consider their addition beneficial. These fits represent robust estimates of smooth developmental trends. Observed means necessarily reflect various sources of noise, as well as ‘true’ developmental effects (i.e. the developmental trajectories that would be observed when sampling an infinite number of participants viewing an infinite number of images). A core assumption underlying the curve fitting is that polynomials capture true and smooth developmental trajectories well but are less perturbed by unreliable bin-to-bin fluctuations than observed averages.

If this assumption is correct, fitted polynomials will be more reliable than observed averages. We now tested this prediction empirically, by comparing the reliability of fitted versus observed trajectories across odd- and even-numbered participants (and across independent odd and even scenes). As shown in the updated Figure S8 (panel A; see below), fitted trajectories generalize extremely well across participant splits (dashed lines) and do so to a greater extent than observed means (bar plots). The same was true across *image* splits (see further down, Figure S9, panel A). This shows that fitted curves are indeed more robust than observed means.

Relatedly, our previous analyses demonstrate that fitted curves are excellent predictors of observed means for hold-out participants (Figure S8 B) and scenes (Figure S9 B).

We think this robustness renders the curve fits a worthy addition to our plots and agree with reviewer 2’s previous assessment of them being a strength. However, we do not consider this central to our study and conclusions and are therefore willing to drop the curve fits in case the reviewer does not find our arguments compelling.

We have now added this cross-validation to p. 4, l. 61-75 of the updated supplementary methods:

Cross-validation of curve fitting across participants and images

One motivation for using curve fitting in the present study is the assumption that curve-fitted trajectories more closely reflect true developmental trends, which we expect to be smooth and replicable, while observed bin-by-bin deviations from smooth trends are likely unreliable. To test this assumption for our present data, we first compared the replicability of fitted and observed trajectories across independent sets of participants. Specifically, we examined trajectories for dwell time and first fixation proportions for four semantic categories (text, faces, touched objects, and bodies), as well as for the centre bias, horizontal bias, visual exploration, and intra-individual entropy. We first identified the most parsimonious model for each developmental trajectory, once using data from all odd-numbered participants and then again using data from all even-numbered participants (see the methods section for a detailed explanation of how the best-fitting model was selected). Then we correlated these fitted trajectories across odd and even participants. Similarly, we calculated the observed bin-wise mean values for each metric separately for odd- and even-numbered participants and correlated those with each other.

Moreover, we tested whether the curve fits for odd-numbered participants generalise to the observed means from even-numbered participants. Following Spearman², we estimated the noise ceiling for correlations between fits and empirical data as the geometric mean of their respective split-half reliabilities (see above).

And p. 15-16, l. 233-258 of the updated Supplementary Results

Cross-validation of curve fitting across participants and images

Figure S8A presents the reliabilities of curve fits (dashed lines) and observed means (bars) across participants. The reliability of curve fits across participants was close to ceiling for dwell time proportions (text: $r(25) = 1$, $p < .001$; faces: $r(25) = 0.87$, $p < .001$; touched: $r(25) = 0.94$, $p < .001$; bodies: $r(25) = 0.99$, $p < .001$), first fixation proportions (text: $r(25) = 0.99$, $p < .001$; faces: $r(25) = 0.96$, $p < .001$; touched: $r(25) = 0.94$, $p < .001$; bodies: $r(25) = 0.97$, $p < .001$), as well as the centre bias ($r(25) = 0.97$, $p < .001$), visual exploration ($r(25) = 0.97$, $p < .001$), intra-individual entropy ($r(25) = 0.96$, $p < .001$) and the horizontal bias ($r(25) = 0.96$, $p < .001$). The reliabilities of observed means were still high, but somewhat lower for all tested metrics (dwell time proportions; text: $r(25) = 0.98$, $p < .001$; faces: $r(25) = 0.76$, $p < .001$; touched: $r(25) = 0.76$, $p < .001$; bodies: $r(25) = 0.43$, $p < .001$), first fixation proportions; text: $r(25) = 0.94$, $p < .001$; faces: $r(25) = 0.86$, $p < .001$; touched: $r(25) = 0.86$, $p < .001$; bodies: $r(25) = 0.57$, $p < .001$), as well as the centre bias ($r(25) = 0.94$, $p < .001$), visual exploration ($r(25) = 0.92$, $p < .001$), intra-individual entropy ($r(25) = 0.91$, $p < .001$) and the horizontal bias ($r(25) = 0.87$, $p < .001$).

Figure S8B shows the generalisation of fitted trajectories for data from odd participants to observed means from even participants, as well as the corresponding estimated noise ceilings. We observed large cross-validated correlations between fitted developmental trajectories and hold-out data for dwell time proportions (text: $r(25) = 0.98$, $p < .001$; faces: $r(25) = 0.81$, $p < .001$; touched: $r(25) = 0.86$, $p < .001$; bodies: $r(25) = 0.61$, $p < .001$), first fixation proportions (text: $r(25) = 0.94$, $p < .001$; faces: $r(25) = 0.87$, $p < .001$; touched: $r(25) = 0.90$, $p < .001$; bodies: $r(25) = 0.66$, $p < .001$), as well as the centre bias ($r(25) = 0.96$, $p < .001$), visual exploration ($r(25) = 0.93$, $p < .001$), intra-individual entropy ($r(25) = 0.94$, $p < .001$) and the horizontal bias ($r(25) = 0.82$, $p < .001$). All correlations were close to or exceeded their estimated noise ceilings (see Figure S8B). Together, this indicates the robustness of the applied curve fitting across participants.

Figure S8. Consistency of curve fits. (A) Dashed lines show Pearson correlations between odd- and even-numbered participants for polynomial curve fits of developmental trajectories. Bar plots show the corresponding correlations without curve fitting (i.e. for observed bin-wise means), which are lower for all metrics. (B) Bar plots show cross-validated Pearson correlations between observed developmental trajectories (bin-wise means of even participants) and polynomial curve fits (fitted to bin-wise means from odd participants). All cross-validated correlations are close to the respective noise ceiling, which is defined as the average of the split-half consistencies shown in (A) and indicated by the horizontal lines. All error bars and error shades indicate standard errors of correlation estimates.

I also checked the author's discussion of the R2 question about Figure S7, where cross-validation was conducted by splitting the dataset in half. The results of this experiment is unreliable, as dividing the data into two equal parts may still retain factors such as the presentation order of stimuli. This similarity between subsets naturally leads to comparable results.

Please note that we conducted separate analyses to probe the robustness of developmental trajectories and fits across independent sets of *participants* (former Fig S7, now S8b) and *images* (former Figs. S8 and S9, now S9). It is correct that the analysis referred to here does probe generalisation across

participants, not images. See Figure S9 and below, for the corresponding cross-validation across images.

More importantly, I find it problematic that the paper's analyses focus exclusively on age as the primary variable. As other reviewers have pointed out, the order of stimulus presentation seems to have a noticeable effect, as shown in Figure S8. Yet, this issue was not considered in the design of the author's other experiments, which seems inappropriate.

This is why, in my initial review, I suggested that the author provide a clear justification for the use of curve fitting. Given that the experimental setup is fixed, e.g, fixed presentation order, the fitted curve only reflects variation within the fixed conditions, rather than offering a generalized result. Furthermore, when the age data consists solely of integers, there seems to be even less need for fitting, as the average values for each age group can be used directly.

It is important to re-iterate that the developmental trajectories we observed generalise well across independent sets of scenes (new Figure S9). Importantly, this also holds for the fitted trajectories, as shown in Panel A; see below

Figure S9. Consistency of curve fits across images. (A) Dashed lines show Pearson correlations between odd- and even scenes for polynomial curve fits of developmental trajectories. Bar plots show the corresponding correlations between observed trajectories (i.e. for observed bin-wise means). (B) Bar plots show Pearson correlations between observed developmental trajectories (bin-wise means of even scenes) and polynomial curve fits (fitted to bin-wise means from odd participants). All correlations are close to the respective noise ceiling, which is defined as the average of the split-half consistencies shown in (A) and indicated by the horizontal lines. All error bars and error shades indicate standard errors of correlation estimates.

We now realise our former Fig. S8 may have given a misleading impression and would like to apologise for that. This figure showed that *individual differences* among adults generalise well between the 40 scenes we used and an independent set of 660 further scenes (panel A, now Fig. S10). Our former Fig S8 also showed that *individual differences* are well correlated across repeated presentations of these 40 scenes in different orders, albeit not perfectly (panel B, now dropped). It

is important to note that this is an extremely conservative test regarding the metric of interest in our manuscript: (age specific) group averages. Averages are much more robust than individual differences. The standard error of the mean (s.e.m.) corresponds to a fraction of the variability at the individual level. Crucially, the data shown in former Fig. S8 also did not allow to distinguish between true effects of stimulus order and simple limits to reliability across repeated tests.

To address this, we now directly tested and compared the re-test error of *group means* across fixed and shuffled stimulus orders. We conducted a new control experiment with $n = 161$ adult participants, who freely viewed the 40 images used in the main study on two separate days, ~ 5 days apart and with the same image order across days. Additionally, we harnessed previous data from $n = 101$ adult participants viewing the stimuli on two separate days, one week apart, but with the stimulus order shuffled on the second day. This also allowed us to compare the magnitude of re-test errors (i.e. differences of group means across testing days) to the size of developmental effects observed in our main sample.

The results of this analysis are shown in the new Fig. S11 (see below). They show that across all key metrics, retest errors of means are dwarfed by the size of observed developmental effects, which are orders of magnitude larger. Crucially, this is true for both, shuffled and constant stimulus orders and there is no consistent effect of stimulus order on re-test consistency. We conclude that stimulus order has no relevant effects on the dependent variables in our design.

We now report on these analyses in the supplementary materials on p. 5-7, l. 89-149

Reliability of means and effects of stimulus order

To test the reliability of group means and investigate a potential image order effect, we empirically determined the re-test error of observed means for adults completing the free-viewing experiment (OSIE 40) twice. For one sample, the stimulus order was constant across testing days, for the other it was shuffled during the re-test.

For the shuffled condition, we re-analysed previously published data (Linka et al., 2020). In that study, $n = 101$ adults participated in a free-viewing experiment conducted over two sessions one week apart. During the first session, participants viewed 700 images, with the initial 40 images being the ones included in our current study. On the second testing day, the same participants viewed 200 images, starting with the same 40 images, now presented in a shuffled order. To compute the re-test error for a given metric, we computed the absolute difference for the respective group mean between testing days.

For the adult sample viewing stimuli in a constant order on both days, we conducted a new experiment and calculated re-test errors in the same way (see below for details on experimental methods).

We then compared re-test errors for shuffled and constant stimulus orders to the observed size of developmental changes in our main sample. Observed developmental changes were defined as the difference between the peak and minimum values of each fitted developmental trajectory (bin-wise mean values; $n = 6720$).

Supplemental stimulus order experiment: Subjects

A total of 397 datasets were collected at Justus Liebig University from $n=223$ participants ($M_{age} = 26.33$, $SD = 8.66$) of whom 137 identified as female (61%), 84 as male (38%), and 2 as diverse (1%). Of these, 18 datasets were excluded due to an average validation error exceeding 1 dva. Additionally, 10 datasets were removed, because they contained incomplete data for more than half of the 40 presented images. Finally, 47 datasets were discarded as participants attended only one session. This resulted in a final sample of 322 datasets from 161 participants, with testing sessions separated by an average of 5.9 days (range: 3 to 13 days). The study received approval from the Local Ethics Committee of the Department of Psychology (Fb06) at Justus Liebig University Giessen. All participants were recruited on campus and provided written informed consent before the study began.

Supplemental stimulus order experiment: Apparatus & stimuli

To closely replicate the setup of the main study, we developed and made use of two mobile eye-tracking stations (see Figure S2 for an image of one setup) that used hardware and software tightly matched to those used in the eye-tracking exhibit for the main study. Eye movements were recorded binocularly using a Tobii Pro Spark Eye Tracker (Tobii AB, Danderyd, Sweden) operating in head-free, remote mode at a distance of 50-90 cm with a sampling rate of 60 Hz. The median validation accuracy for this study was 0.39 dva. ($SD = 1$).

Figure S2. Eyetracking station set up. Image of one of the two eyetracking stations used for collecting eyetracking data. To facilitate consistent reassignment of participants to the same station across testing sessions, the stations were labeled with distinct names: 'Lise

Supplemental stimulus order experiment: Procedure

The procedure closely followed that of the main experiment. The main difference was that participants entered a personalised ID, which enabled matching of datasets across testing days. Also, participants received instructions in person rather than via a pre-recorded video. Participants sat at an average distance of 63 cm from the screen. After completing the first session, participants received a document detailing the earliest date (minimum of 3 days later) they should return to for their second testing session. The presentation order of images was consistent across all participants and between testing sessions. After completing the second testing session, participants were compensated with 10 euros.

Data processing

All pre-processing and statistical analyses were identical to the main experiment.

and p. 21-22, l. 311-326

Reliability of means and effects of stimulus order

To test the reliability of group means and a potential effect of image order, we compared the observed developmental change for each key metric in our main sample to the re-test error from adults completing the free viewing task twice, with either fixed or shuffled stimulus order during the second session. Figure

S11 below shows developmental changes and retest errors for each metric and condition.

Our results show that retest errors for all tested metrics, which showed a developmental effect are orders of magnitude smaller than corresponding developmental changes. The following values indicate developmental changes (and re-test errors for fixed and shuffled image orders in brackets, separated with a comma): Dwell time proportions for text: 14.10 (1.5, 0.53); faces: 6.8 (0.62, 0.28); touched objects: 3.2 (0.07, 0.86); bodies: 1.97 (0.49, 0.31); First fixation proportions for text: 8.52 (0.63, 0.01); faces: 10.64 (1.7, 2.33); touched objects: 6.41 (1.94, 1.84); bodies: 2.46 (0.93, 2.45); Centre bias: 41.38 (0.52, 1.49); visual exploration: 40.94 (4.66, 5.63); entropy: 0.96 (0.02,

Figure S11. Developmental change and retest error for key metrics. Lines represent developmental change, calculated as the difference between the peak and minimum values of a given fitted developmental trajectory. Filled dots and circles represent retest errors for shuffled and constant image orders, respectively. Retest errors were calculated as the absolute between-session difference of respective means for adult samples (see text for details). (A) shows developmental changes and re-test errors for average dwell time- and first fixation proportions across semantic dimensions, (B) for other key metrics as indicated below the x-axes.

0.02); horizontal bias: 4.45 (0.23, 0.12). Importantly, this was true independent of image order and there was no consistent effect of constant or shuffled image orders on re-test errors.

We further discuss these findings on p. 21, l. 409-416.

To further validate the robustness of our findings, we compared the size of developmental changes for each trajectory against the retest errors observed for corresponding means in independently acquired adult data. Observers in these studies took part in two free-viewing sessions using the same stimuli used in our main experiment. The image order was either fixed or shuffled across sessions. The results of these experiments showed retest errors which were orders of magnitude smaller than the observed developmental effects in the main sample, regardless of whether the stimulus order was fixed or shuffled between sessions (Supplementary Materials; pp. 21-22; Figure S12).

Taken together, all control analyses underscore the generalizability of our findings across participants, scenes and image orders. Importantly, across participants and scenes, fitted curves proved more robust than observed averages. Therefore, we consider the addition of curve fits worthwhile and argue for keeping them. However, as mentioned above, we are open to dropping the fits in case the reviewer does not find our arguments compelling.

Minor: I observed in Figure S2 that there are significant deviation in gaze accuracy error across different age groups, with notable variability for children aged 5 to 6. I wonder whether this could introduce bias into the analysis of younger children.

We appreciate the reviewer's comment. Across all *collected* datasets, we did indeed observe an increased median validation error of 0.78 dva for children aged 5-6. However, all participants with an average validation error above 1 dva were excluded, as previously reported. Among those included, the median validation error for this age group was 0.54 dva, closely aligned with the median validation error across all age groups (0.47 dva).

Reviewer #2 (Remarks to the Author):

I would like to express my thanks for the authors response to the reviewers' comments and my satisfaction with the provided thorough evaluation and manuscript modifications regarding the addressed concerns. To improve the manuscript for publication, I would add up the following minor comments:

- page 2 line 28: The authors state previous evidence points out 'mature' viewing behaviour from about 8 years of age. However, I find the word 'mature' in the context misleading and uncommon in the literature in connection with the mentioned age. Could the authors provide reference on common usage of this term in the literature, or alter the statement to be conform with the consensus?

We thank the reviewer for pointing this out. We have now changed the wording to 'adult-like' throughout the manuscript. For example, see page 2, lines 27-28.

*Previous evidence is limited and suggests viewing behaviour may be **adult-like** from about 8 years of age*

- pages 4-5 lines 96,111: The authors use two terms to reference the target sample of their research: 'adolescence' and 'teenage'. While these are generally understandable to the audience, I would recommend to unify the terms in the manuscript, unless the authors introduce both these terms intentionally and refer to various target samples.

We thank the reviewer for pointing this out. We have now used the term 'adolescence' consistently throughout the manuscript. For example, see p. 5, l. 115-116.

*We find strong evidence for a protracted development of attentional preferences and basic oculomotor biases extending into **adolescence** and young adulthood*

- page 8 lines 147-149: "Overall, the developmental trajectory of dwell time proportions was best fit by a 3rd degree polynomial, while that of first fixations was best fit by a linear model with a slight positive slope." - I would recommend the authors reference the methodology to find the best fit here, and in the appropriate occurrences of another such assumptions in the manuscript. Therefore, the reader has all the information needed to understand the authors' claims always directly in the place.

We thank the reviewer for the recommendation. We have now added a description of how we selected the best fit at the beginning of the results section (p. 5, l. 124-128):

Specifically, we fitted polynomials of degrees ranging from 0 to 10 to the data for each metric. For these candidate models, we calculated the AIC differences using the formula $\Delta i = AIC_i - AIC_{min}$, where AIC_i is the AIC value of the i -th model, and AIC_{min} is the lowest AIC value among all models. We then selected the least complex model with $\Delta i < 4$ as the best fit.

And reference this at the beginning of each paragraph of a given analysis. (See p. 10, l. 179-182 for an example):

*The proportion of horizontal saccades increased from 53% to 56% from ages 5-6 to at least 13-14 ($\Delta AIC > 5$) and the best fitting developmental trajectory (9th degree polynomial) reached a plateau from about age 15-16 (*see above for details on model selection*).*

- I highly appreciate introducing results regarding the protracted development of gender differences and I find the results and corresponding discussion interesting. I would recommend that the manuscript points out the state-of-the-art in this particular domain earlier in the introduction and incorporates the proposed evaluation objectives into the story of the paper contribution in the introductory parts because the interconnection of the provided evaluation with the aims of the manuscript is now missing.

We thank the reviewer for pointing this out. We have now included the relevant literature into the introduction on p. 3-4, l. 76-79.

Further, while adult females show enhanced processing of some social stimuli⁴¹⁻⁴³, this does not seem to translate to generally enhanced face

salience⁴⁴ and it is unclear whether there are relevant potential gender differences across development

and mentioned the analyses on gender differences as part of our objectives on p. 4, l. 100-102.

The size of the present dataset further allowed us to investigate gender differences in semantic salience and their potential variation across age

- Figure 1C: bottom-left graph contains blue-red marks without legend; the evidence of change in the visualizations is hardly visible except the strongest evidence, moreover gray patches conflict with the colour scale used for the evidence of change

We completely agree and have now updated the legends to Figure 1C (see below). Additionally, we have revised all figures to move grey bars indicating evidence of linear change to the top and plotting them in more distinct shades of grey. Furthermore, we have changed the colour of the patches to red. Please see below for an example.

A

B

C

D

$\Delta AIC > 25$ $\Delta AIC > 15$ $\Delta AIC > 5$
 Evidence for change

◆ / ● Age 5-6
 ◆ / ● Age 19-20

E

Figure 1. Protracted development of semantic salience. Scatter points display the mean proportion of dwell time and first fixations directed towards objects of four semantic features across age bins displayed on one axis (A-B) and zoomed in on each semantic category separately (C-D). Please note that each age bin spans two years. Due to space constraints, only every second tick is labelled, and these are marked in bold for easier identification. Semantic categories are indicated by colours, as shown in the inset. Error bars indicate ± 1 standard error of the mean (SEM). Lines show the corresponding best fitting polynomials. The horizontal lines at the top of a given figure indicate the level of evidence for a given bin to be part of broad linear change via shades of grey, as shown in the inset (sliding window analysis, cf. main text and methods). Red patches mark age bins with lower sensitivity to developmental changes, which were disregarded for curve fitting ($n < 90$ per bin). The left-hand side of panel (E) displays object pixel masks for an example image, with colors corresponding to categories in A - D (cf. inset, objects outside these categories shown in light gray). The middle and right-hand images display alpha-blended fixation overlays for the same image, showing all fixations (middle) or only first fixations after image onset (right-hand side) for two age groups. Fixation overlays shown in blue and red correspond to ages 5-6 and 19-20, respectively. These age groups are marked correspondingly with blue and red rhombuses above the x-axes in Panels (C) and (D).

- Figure 2B, 3A-C: This Figure should contain more sophisticated visual attributes mapping, so the trendline can be visually better comparable to the observed trends (i.e. by using colour attributes).

We thank the reviewer for bringing this to our attention. We now plot observed averages as scatters, rendering the observed means and fitted trends clearly discernible.

- I would recommend the authors to thoroughly proof-read the paper before submission for the publication (see inconsistency in spelling of the word 'behavior' in page 19, line 353, etc.)

We thank the reviewer for making us aware of that. We've carefully proofread the manuscript and standardised the spelling to British English, including the spelling of 'behaviour'.

Reviewer #2 (Remarks on code availability):

I would encourage the authors to provide the raw eyetracking data publicly available alongside with the publication of the manuscript to elevate the contribution of their research (future raw data publication promise in readme file).

Of course, we have updated all data from each processing stage, including the code for all analyses presented in the manuscript, on our OSF project page

Reviewer #3 (Remarks to the Author):

The authors have made considerable effort to improve the manuscript. I was satisfied with the answers to my questions and the resulting revisions. I find the paper ready for publication.

We thank all reviewers for their help in getting the paper to this stage!

Dear Dr. [REDACTED]

On behalf of all authors, I'd like to sincerely thank you and the reviewers for your valuable feedback and for helping bring our manuscript to this stage. We very much appreciate all the helpful comments and suggestions.

As requested, we've submitted all formatted documents according to the provided checklist. Below, you'll also find a brief summary of our main findings for social media and web presence purposes. Please let us know if anything is missing or needs any adjustments.

Summary

Linka and colleagues recorded eye movements of thousands of children and adults viewing scene images in a museum. Adult gaze was marked by well-established spatial and semantic biases. Surprisingly, children deviated from this well into teenage

Best regards,
Marcel Linka (on behalf of all authors)